

**Projected climate change will double the Late Holocene maximum to present ice**
**loss in Central-Western Greenland by 2070**
Josep Bonsoms [1*], Marc Oliva [1], Juan Ignacio López-Moreno [2], Guillaume Jouvet [3]
[1*] Department of Geography, Universitat de Barcelona, Spain.
Email address: josepbonsoms5@ub.edu
[2] Instituto Pirenaico de Ecología (IPE-CSIC), Campus de Aula Dei, Zaragoza, Spain.
[3] Institute of Earth Surface Dynamics, University of Lausanne, Lausanne, Switzerland.
Greenland's peripheral glaciers and ice caps (GICs) have experienced accelerated mass
loss since the 1990s. However, the extent to which present and future trends of GICs are
unprecedented within the Holocene is poorly understood. This study bridges the gap
between the maximum ice extent (MIE) of the Late Holocene, present and future glacier
evolution until 2100 in Eastern Nuussuaq Peninsula (Central-Western Greenland), where
the age of moraine boulders was determined by surface exposure dating. The Instructed
Glacier Model (IGM) is calibrated and validated by simulating present-day glacier area
and ice thickness. The model is employed to reconstruct eastern Nuussuaq Peninsula
GICs to align with the MIE of the Late Holocene, which occurred during the late
Medieval Warm Period (1130 ± 40 and 925 ± 80 CE). Subsequently, the model is forced
with CMIP6 projections for SSP2-4.5 and SSP5-8.5 scenarios (2020-2100). Glaciers
reach the MIE of the Late Holocene when temperatures decrease between 0.75ºC and 1ºC
relative to the baseline climate period (1960-1990). Currently, glaciers have retreated by
34% compared to the MIE of the Late Holocene. By the end of the 21st century (2100),
temperatures are projected rise up to 6ºC (SSP5-8.5) with respect to the baseline climate,
exceeding temperatures prevailing during the Holocene Warm Period (~10 to 6 ka) by a
factor of three. Using IGM with a positive degree-day model calibrated with geodetic
mass balance data from 2000-2020, we project that by >2070 under SSP2-4.5 and SSP5-
8.5, glacier mass loss will double (-70%) the loss trend observed from the MIE of the
Late Holocene to the present. This work helps contextualize present and future glacier
retreat within a geologic time scale and quantify the impacts of anthropogenic climate
change on the cryosphere.
**Key words:** Climate change, glaciology, glacial geomorphology, numerical modelling,
deglaciation, Greenland, Arctic.
**1. Introduction**
Arctic temperatures are rising at a faster (~ 4 times) rate than the global average (IPCC,
2019) and glaciers are accelerating the mass loss (Hugonnet et al., 2021). In 2021,
Greenland's peripheral glaciers and ice caps (GICs) represented a small (4%) ice cover
area of the island but contributed to 11% of the total Greenland ice loss and sea level rise
(Khan et al., 2022). The recession of glaciers implies alterations in fauna and flora



patterns (Saros et al., 2019), as well as impacts on water availability, climate, and ocean
and atmospheric dynamics that have environmental and climate consequences far beyond
the polar regions (IPCC, 2022).
The Little Ice Age (LIA; 1300-1900 CE) has been defined as the last period with
widespread glacier expansion (Kjær et al., 2022). Greenland GICs have lost 499 Gt of ice
from end of LIA to 2021 (Carrivick et al., 2023). The rate of loss of GICs has increased
since the 1990s (Bölch et al., 2013; Larocca et al., 2023), with recent trends indicating an
acceleration in mass loss from $27.2 \pm 6.2$ Gt/yr (February 2003–October 2009) to $42.3 \pm$
$6.2$ Gt/yr (October 2018–December 2021) (Khan et al., 2022). Warming rates have been
higher in West Greenland than in the East since the end of the LIA (Hanna et al., 2012).
As a result, the loss of ice from Greenland's GICs has been more pronounced in its western
fringe, where warmer conditions have been associated with the positive phase of the
North Atlantic Oscillation (NAO), resulting in a West-to-East warming gradient (Bjørk et
al., 2018). Historical records reveal varying trends in GICs over the past two centuries.
Aerial images and satellite data indicate that GICs in the West Greenland remained
relatively stable, maintaining their extent from the mid-19th century until the mid-20[th]
century, after which they experienced rapid retreat (Weidick, 1994; Leclerq et al., 2012).
For instance, Citterio et al. (2009) observed a reduction in glacier area of approximately
20% from the LIA to 2001. Other estimates suggest a 48% loss in GICs area in Southern-
Western Greenland since the maximum extent of LIA up to 2019 (Brooks et al., 2022).
The recent evolution of the GICs has been reconstructed using historical aerial images
and satellite records (Leclerq et al., 2012; Yde and Knudsen, 2007; Citterio et al., 2009;
Bjørk et al., 2018; Larocca et al., 2023). Geospatial techniques, such as the inference of
the Equilibrium Line Altitude (ELA), have also been utilized (Brooks et al., 2022;
Carrivick et al., 2023). However, aerial and satellite images provide temporal data over
centuries and decades and geospatial methods neglect ice-flow physics and do not account
for glacier dynamics. Based on the distribution of moraines and unvegetated trimlines in
Central-Western Greenland, some authors suggested that the Late Holocene maximum
glacier extent occurred around the LIA (Humlum, 1999). However, cosmic ray exposure
(CRE) dating of erosive and depositional glacial records indicates that the maximum ice
extent (MIE) of the Late Holocene did not occur during the LIA in many areas in Western
Greenland but during the Medieval Warm Period (MWP; 950 to 1250 CE) (Young et al.,
2015; Jomelli et al., 2016; Schweinsberg et al., 2019).
Evidence from physical-based modelling of the GICs recession during the Holocene
remains limited compared to those near the GrIS (i.e., Cuzzone et al., 2019; Briner et al.,
2020). Holocene reconstructions of GrIS extent based on physical modelling, guided by
geomorphological evidence, provide valuable insights into the paleoclimate conditions
that led to the MIE of the Late Holocene and subsequent recession (Simpson et al., 2009;
Lecavalier et al., 2014; Cuzzone et al., 2019), facilitating comparisons of past and future
glacier responses to climate change (Briner et al., 2020). Physical-based ice-flow
modelling relying on full-Stokes equations are computationally intensive at high



resolution (sub-kilometer) for long-term paleo glacier simulations and model parameter
calibrations (Jouvet et al., 2022). Simplified models such as the hydrostatic Shallow Ice
Approximation (SIA) and the Shallow Shelf Approximation (SSA) tend to overestimate
ice velocities near glacier margins and underestimates velocities in deep glaciated areas,
respectively. An emulator based on a convolutional neural network (CNN), trained with
high-order ice flow equations, offers reduced computational costs while maintaining
accurate ice thickness estimates comparable to those obtained through high-order
equations (Jouvet, 2023a).
The future recession of Greenland GICs compared to the long-term Holocene fluctuations
is poorly understood. Here, we calibrate and validate the Instructed Glacier Model (IGM)
(Jouvet et al., 2023a), a glacier evolution model based on a CNN emulator to estimate ice
flow, to reconstruct the MIE of the Late Holocene in an extended glacier area in the
Eastern Nuussuaq Peninsula (Central-Western Greenland). This area has CRE records
available for the outermost glacier moraine complexes but the paleoclimate conditions
causing these glacier oscillations are not yet known in detail (D'andrea et al., 2011; Biette
et al., 2019; Jomelli et al., 2016; Schweinsberg et al., 2019; Osman et al., 2021).
Employing IGM allows us to reconstruct glaciers in high (90 m) resolution based on high-
order equations (Jouvet, 2023a), demonstrating the methodology's capabilities for glacier
modeling at regional scales. Future glacier evolution is modeled under the CMIP6 SSP2-
4.5 and SSP5-8.5 scenarios, from present and steady-state glacier conditions to the year
2100. We compared the projected ice loss trend against the reconstructed MIE of the Late
Holocene to the present-day ice loss trends, extending glacier records from decades to
millennia and placing present and future glacier shrinkage within a long-term Holocene
perspective.
The objectives of this work are to (i) reconstruct past glaciers under different climate
conditions, (ii) determine past and future climate conditions influencing the MIE of the
Late Holocene and future glacier recession, (iii) quantify future glacier retreat trends, and
(iv) compare future ice loss trends with the rate of ice loss from the MIE of the Late
Holocene to the present.
**2. Study area**
This study focuses on a land-terminating glacier area in the Nuussuaq Peninsula, Central-
West Greenland (Figure 1). This peninsula extends from the onshore Disko (South) to
Svartenhuk Halvo (North). Nuussuaq Peninsula includes several mountain glaciers and
ice caps connected to the GrIS that surrounds its eastern flank. Our study focuses on a
glacier area in the Eastern Nuussuaq Peninsula, with elevations ranging from 400 to 1200
meters above sea level (m a.s.l.) (Figure 1).
Present-day climate conditions are characterized by a polar maritime climate, becoming
more continental toward the inland areas and GrIS (Humlum, 1999). Moist air masses
from the Davis Strait influence the climate during summer, with continental polar air
influences during the winter (Ingolfsson et al., 1990). Prevailing winds in the region



typically come from the East and North-East, except during the summer months, when
Southerly and Southern-Western winds prevail (Humlum, 1999). The relief configuration
exposes Disko Bugt to cyclogenic activity and moist airflow, resulting in decreased
precipitation from the peripheral coastal areas towards the GrIS (Weidick and Bennike,
2007). The nearest research station with meteorological and snow observations is the
Arctic station, at coastal Disko Island (Central-Western Greenland). Here, the
accumulated annual precipitation is 436 mm (1991–2004 period) (Hansen et al., 2006).
The mean annual temperature (MAAT) is -4°C (1961–1990 period), with a lapse rate of
around 0.6°C per 100 m (Humlum, 1998). At Arctic station, the snow season typically
extends from September to June, with maximum snow accumulations of around 50 cm
(Bonsoms et al., 2024).
The present-day landscape in Central-West Greenland is characterized by the presence of
glaciers, which have also intensely shaped the relief in ice-free areas in the past. Today,
environmental dynamics in these areas is strongly influenced by periglacial processes
under a continuous permafrost regime (Humlum, 1998; Christiansen et al., 2010) that
reshape the geological setting made of clastic sediments from the Mid-Cretaceous to the
Palaeogene (Pedersen et al., 2002). The strong glacial imprint in the landscape of the
peninsula results from a complex glacial history, which is not yet known in detail.
Following the LGM, the GrIS underwent a significant retreat during Termination-1 and
exposed the coastal regions in Central-West Greenland (Briner et al., 2020). As in other
regions across Greenland, the Early Holocene was characterized by warm temperatures
that led glaciers to retreat (Leger et al., 2024). In the Nuussuaq Peninsula, CRE records
reported the onset of glacial retreat by ca. 10 ka (O'Hara et al., 2017). The minimum GrIS
extension occurred from ca. 5 to 3 ka cal BP, when GrIS margins retreated by ca. 150 km
from present-day terminus position (Briner et al., 2016), which explains the lack of glacial
records corresponding to the Early-Mid Holocene in the peninsula (Kelly and Lowell,
2009; O'Hara et al., 2017). According to several absolute dating methods in different
natural records, the Nuussuaq Peninsula GICs grew between approximately 4.3 and 2 ka
and reached several glacier culminations during the past millennium before the LIA
(Schweinsberg et al., 2017; 2019). The internal and external moraine complexes in the
area reported CRE ages of $1130 \pm 40$ and $925 \pm 80$ CE, respectively (Young et al., 2015).
These ages are consistent with other CRE ages obtained in Central-Western Greenland
for the most external recent moraine complexes, indicating that the late MWP glacier
expansion was the largest of the Late Holocene (Jomelli et al., 2016; Schweinsberg et al.,
154   2019)





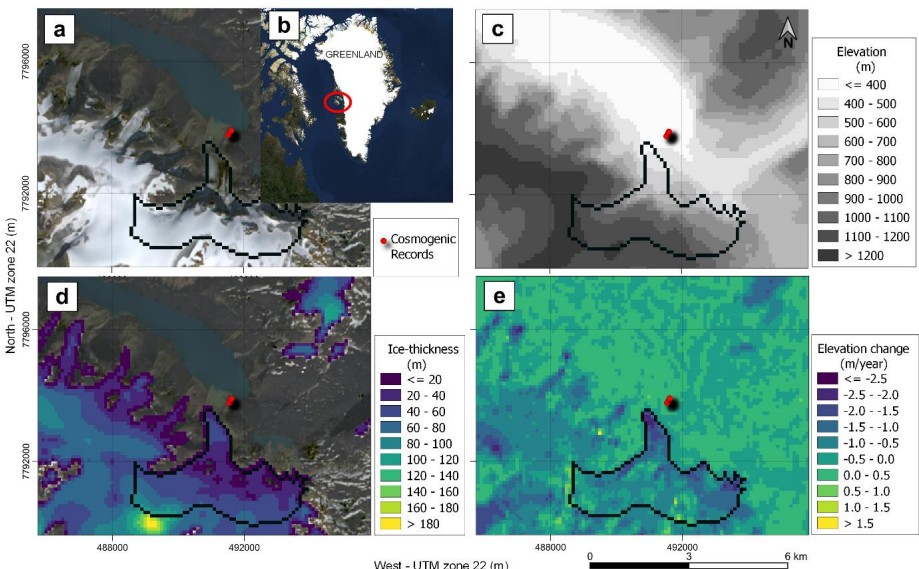

**Figure 1.** Location of the reconstructed glacier and CRE ages (red points) used in this work (a). Location of the study area within Greenland (b). Glacier delimitation is based on Randolph Glacier Inventory (RGI6). The base map is a Sentinel-2 image from https://s2maps.eu/ (2022). Elevation map of the study area from a Digital Elevation Model Copernicus DEM GLO-90 (2010-2015) (c). Average ice thickness (m) (2018-2022) from Millan et al. (2022) (d). Elevation changes average values (m/year) between 2000-2019 (Hugonnet et al., 2021) (e).

## 3. Data

### 3.1 Pre-processing glacier data downloading

The data used to force and validate the IGM is detailed at Table 1. Topography data were obtained from a Copernicus Digital Elevation Model (DEM) with a resolution of 90 meters (COPERNICUS DEM GLO-90). In-situ mass balance records in the study area are scarce and fieldwork is challenging. However, recent advancements in global-scale mass balance, satellite imagery, and ice thickness estimates have enabled the validation of the glacier model. We compared the ice thickness estimates from Farinotti et al. (2019), which are the output from an ensemble of five models (HF-model, GlabTop2, OGGM, GlabTop2 IITB version, and an unnamed model), with those from Millan et al. (2022), which are derived from numerical modeling based on SIA and data obtained from a constellation of remote sensing products (Sentinel-1/ESA, Sentinel-2/ESA, Landsat-8/USGS, Venµs/CNES-ISA, Pléiades/Airbus D&S). Glacier mask outlines were acquired from the Randolph Glacier Inventory Version 6 (RGI6.0). Elevation change rate (dh/dt) data were obtained from Huggonet et al. (2021).



The climate variables required to run the IGM model are monthly accumulated
precipitation (kg m$^{-2}$ yr$^{-1}$), monthly average air temperature (ºC), and monthly air
temperature standard deviation (ºC) from the nearest pixel to the glacier. We utilized the
GSWP3- W5E5 monthly dataset at a spatial resolution of 0.5° x 0.5°, which combines the
Global Soil Wetness Project phase 3 dataset with the bias-adjusted ERA5 reanalysis
dataset (Cucchi et al., 2020). Future glacier changes are modeled based on bias-corrected
monthly accumulated precipitation and monthly average air temperature CMIP6 multi-
model mean (n=33) for SSP2-4.5 and SSP5-8.5 (2020 to 2100) at a spatial resolution of
0.25° (Thrasher et al., 2022), subtracted at the nearest grid point of the glaciers. Months
are aggregated into seasons as follows: September, October, November (Autumn), March,
April and May (Spring), December, January and February (Winter), June, July and August
(Summer). Data were downloaded using Open Global Glacier Model (OGGM) model
(https://oggm.org/) (Maussion et al., 2015) shop module of IGM (Jouvet et al., 2023),
except for CMIP6 projections (Thrasher et al., 2022) and ice thickness estimates (Farinotti
et al., 2019).
**Table 1.** Characteristics of the datasets employed for forcing, calibrating, and validating
194                                                            the IGM.

| Description | Name | Spatial resolution | Database Date | Source |
|---|---|---|---|---|
| DEM | Copernicus DEM GLO-90 | 90 m | 2010-2015 | https://spacedata.copernicus.eu/documents/20126/0/CSCDA_ESA_User_Licence_2021_11_17.pdf |
| Ice-thickness (1) | Millan et al. (2022) | 100 m | 2017-2018 | Millan et al. (2022) |
| Ice-thickness (2) | Farinotti et al. (2019) | 25 m | 2019 | Farinotti et al. (2019) |
| Baseline climate data | GSWP3_W5E5v2.0 | 0.5° x 0.5° | 1960-1990 | https://data.isimip.org/search/simulation_round/ISIMIP2a/product/InputData/climate_forcing/gswp3-w5e5/ |
| CMIP6 projections | CMIP 6 | 0.25º | 1960-2100 | Thrasher et al. (2022) |
| Dh/dt | Hugonnet et al. (2021) | Glacier (RGI6.0) level | 2000-2020 | Huggonet et al. (2021) |
| Glacier Outline | RGI6.0 | Glacier (RGI6.0) level | 2003 | https://www.glims.org/RGI/ |


**3.2 Geomorphological and paleoclimate data**



The CRE ages are based on nuclide ([10]Be) introduced by Young et al. (2015) and refer to
the period of the maximum glacier advance of the last warm/cold cycles in the Nuussuaq
Peninsula and were used for the paleoclimate modelling purposes of this study. The
sampled boulders were obtained from the outer ridge of the moraine and reveal either (i)
a period of glacial surge or (ii) a phase of stabilization/stillness during the long-term
retreat. However, special caution must be taken when interpreting these ages, as they are
not directly indicative of the period of ice occupation but of the timing of stabilization of
moraine boulders.
Paleoclimate anomalies with respect to the baseline climate were obtained from annual
air temperature reconstructions from ice cores of the GrIS and margins of the GrIS
provided by Buizert et al. (2018). This data ranges from the Last Glacial Maximum
(LGM; ~ 26-19 ka ago) to 2000 CE.
**4. Methods**
**4.1 Instructed Glacier Model (IGM)**
The IGM is a glacier model that simulates ice thickness evolution according to ice mass
conservation principles, surface mass balance and ice flow physics (Jouvet et al., 2023a).
IGM updates the ice thickness at each time step from ice flow and surface mass balance
(SMB) by solving the mass conservation equation. The ice flow is modelled using a CNN
model that is trained to satisfy high-order ice flow equations. There are two main
parameters that control the strength of the ice flow: the Arrhenius factor (A) that controls
the ice viscosity in Glen's flow law (Glen, 1955) and the basal sliding coefficient (c), by
the nonlinear sliding Weertman's law (Weertman, 1957).
Temperature data is downscaled over the DEM using a reference height and a constant
lapse rate of -0.6ºC/100 m, while precipitation is downscaled using a vertical gradient of
35 mm/100 m. Precipitation is classified as solid (< 0ºC) or liquid (> 2ºC), with a linear
transition between solid and liquid phases. The melting threshold is set to -1ºC, and the
density of water is fixed at 1000 kg $^{m-3}$. The SMB is estimated using a monthly positive
degree-day (PDD) model (Hock, 2003; Huss, 2008). The PDD is calibrated based on the
OGGM v1.6.1 SMB calibration process, which is included in IGM SMB module.  OGGM
v1.6.1 SMB calibration correct temperature and precipitation biases from climate data
and adjust the melt factor (5, in this case) to fit the average glacier geodetic mass balance
from January 2000 to January 2020 from Hugonnet et al. (2021). Further details of the
OGGM v1.6.1 SMB calibration process are provided in the OGGM documentation
([https://oggm.org/tutorials/master/notebooks/tutorials/massbalance_global_params.html](https://oggm.org/tutorials/master/notebooks/tutorials/massbalance_global_params.html)
), whereas the physical basis of IGM is detailed in Jouvet et al. (2022; 2023a).
**4.2 Present day glacier calibration and validation**
We calibrated the IGM to simulate RGI6.0 area, and ice thickness from available datasets
(Farinotti et al., 2019; Millan et al., 2022). The IGM parametrization is performed based
on conducting a sensitivity analysis to A and *c*. These parameters were chosen to optimize





IGM and accurately simulate different ice conditions, basal sliding conditions and
subglacial hydrology. An ensemble of IGM parameter options was performed over a
model run of 1000-years with different temperature perturbations of -0.75ºC, -0.5ºC, 0ºC
and +0.25 ºC with respect to baseline climate (1960-1990) in order to reach long-term (>
500 years) glacier area steady-state conditions. The range of temperature perturbation was
determined through trial and error, which showed that values outside this range of
temperature anomalies produced higher discrepancies with respect to the available
datasets used for results validation (Figure 3 to 5). A sensitivity analysis was performed
on IGM parametrization to simulate cold, temperate, and soft ice conditions by changing
A from 34 MPa$^{-3}$ a$^{-1}$, 78 MPa$^{-3}$ a$^{-1}$ (IGM default value) to 150 MPa$^{-3}$ a$^{-1}$. Sliding
conditions are parametrized by changing $c$ from 0.01 km MPa$^{-3}$ a$^{-1}$, 0.03 km MPa$^{-3}$ a$^{-1}$
(IGM default value), and 0.05 km MPa$^{-3}$ a$^{-1}$. The IGM parametrization is shown in
Figures 3 to 5. The remaining parameters were set to the default configuration of the IGM.
The accuracy evaluation of the modeled IGM outputs is based on both area and ice
thickness. We calculated (i) the Mean Absolute Error (MAE) between the accumulated
glacier ice thickness from Farinotti et al. (2019) and Millan et al. (2022) and the output
from IGM; (ii) the glacier area difference between RGI6.0 area and from IGM. To
incorporate both area and ice thickness errors, we calculated the bias by multiplying the
ice thickness MAE (i) by the area difference (ii).

## 4.3 Past and future glacier evolution

IGM is forced with the lowest error parameterization option until the glacier area reaches
present-day and long-term stable-state conditions. The model is run again 1000-years
with an ensemble of different temperature and precipitation values to simulate MIE of the
Late Holocene from MWP. The temperature was perturbed over the baseline climate from
0 to -1ºC by steps of 0.25ºC. Precipitation was non-changed (0%) and increased (10%) in
order to estimate if high rates of snowfall could compensate warming. MIE of the Late
Holocene paleoclimate conditions were determined by calculating the distance between
the glacier tongue of the ensemble of simulations and the CRE dates of the outer ridge
moraines (Köse et al., 2022). The simulations that match the outer ridge moraines
represent the climate conditions before the CRE dates. The present-day glacier area with
steady-state conditions is the starting point of the future simulations (Zekollari et al.,
2019). Subsequently, the IGM is run from the present day until 2100 using monthly
accumulated precipitation and average air temperature CMIP6 multi-model mean SSP2-
4.5 and SSP5-8.5 anomalies with respect to the baseline climate, applying additive factors
for temperature and multiplicative factors for precipitation (Rounce et al., 2023). Present
and future ice thickness anomalies with respect to the MIE of the Late Holocene are
calculated by subtracting the difference between the accumulated ice thickness for the
MIE of the Late Holocene (i) from the accumulated ice thickness from the present-day
(ii) and future ice-loss (iii), dividing by the accumulated ice thickness for the MIE of the
Late Holocene (i), and multiplying by 100. The factor of increase under future climate





change is calculated by dividing future ice loss anomalies by the present-day ice loss
anomalies relative to the MIE of the Late Holocene.

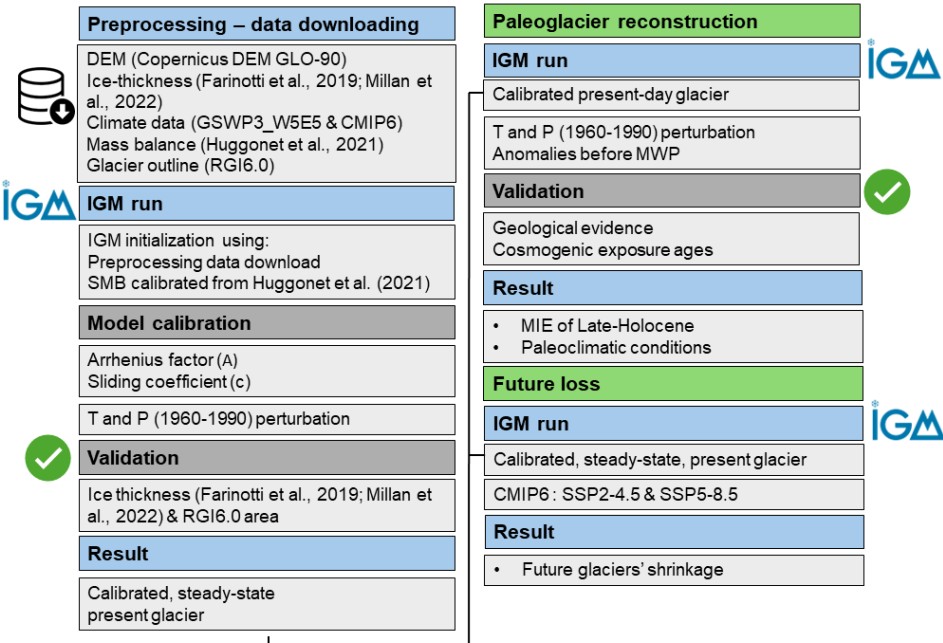


**Figure 2.** Flowchart followed for reconstructing past and present glaciers and projecting
their future evolution based on air temperature (T) and precipitation (P).
**5. Results**
**5.1 IGM parametrization and calibration**
For most IGM parametrizations, glacier growth occurred until 350 to 600 years of spin-
up. The latest year of the spin-up simulation is subsequently validated against ice
thickness estimates (Farinotti et al., 2019; Millan et al., 2022). The error metric values for
ice thickness and area resulting from the IGM calibration and parametrization process are
shown in Figure 3 to 5. The most favorable range for achieving accurate results for
present-day glaciers is a perturbation range of temperature from 0ºC to -0.5ºC with respect
to the baseline climate (Figure 3 to 5). The largest errors in ice thickness and glacier area
were observed for the A = 34 MPa$^{-3}$ a$^{-1}$ and $c$ = 0.01 km MPa$^{-3}$ a$^{-1}$ IGM configuration.
This configuration tended to overestimate ice thickness for both global-scale ice thickness
references (Figure 4 and 5). Additionally, using the default configuration and reducing
the temperature to < -0.5 ºC over the baseline climate led to overestimations of ice
thickness.



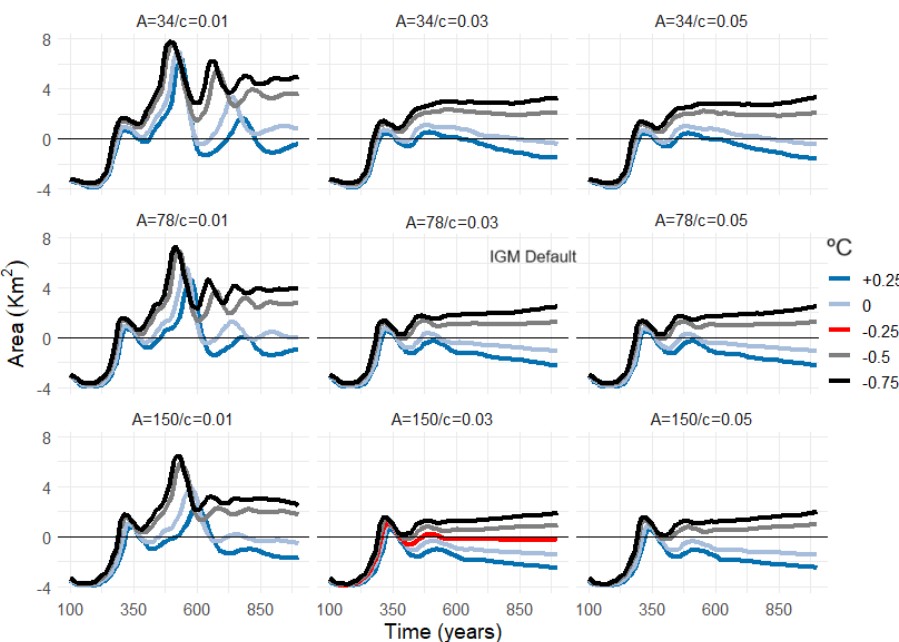

**Figure 3.** Difference from the RGI6.0 area and IGM outputs within a 1000-years spin-up. Data is grouped by changes in temperature (colors), A and *c* options (boxes). The selected configuration (A = 150 MPa$^{-3}$ a$^{-1}$ and *c* = 0.03 km MPa$^{-3}$ a$^{-1}$, -0.25ºC with respect to the baseline climate) is shown in red color.





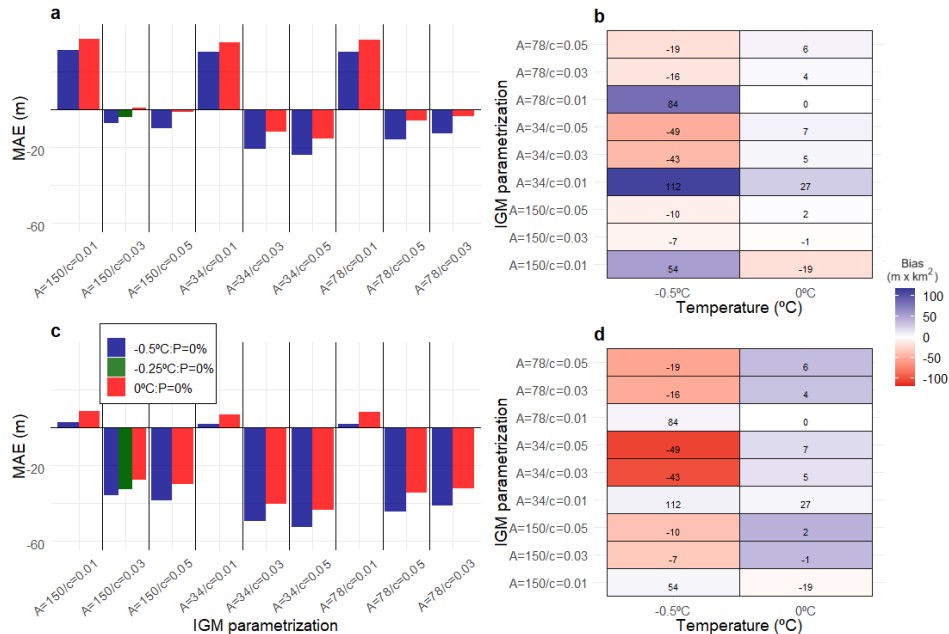

**Figure 4.** Ice thickness MAE between Farinotti et al. (2019) and IGM outputs after spin-up with different A and $c$ parametrizations and perturbations of temperature (a). The selected configuration is shown with green color. Ice thickness MAE values from Figure 4 (a) multiplied by the difference between the RGI6.0 area and the IGM outputs (bias) for different A and $c$ parametrizations and perturbations of temperature (b). Figure 4 (c) and (d) are the same as Figure 4 (a) and (b), respectively, but for ice thickness estimates from Millan et al. (2022).

Trial and error parametrizations of A and c revealed that optimal results were achieved for A = 150 MPa$^{-3}$a$^{-1}$ and $c$ = 0.03 km MPa$^{-3}$ a$^{-1}$. These outputs of the IGM align with Farinotti et al. (2019). However, both IGM ice thickness and Farinotti et al. (2019) overestimate ice thickness compared to Millan et al. (2022) (Figures 4 and 5). Setting A = 150 MPa$^{-3}$a$^{-1}$ and $c$ = 0.03 km MPa$^{-3}$ a$^{-1}$, with a slight variation of temperature (-0.25 ºC) over the baseline climate, resulted in very similar accumulated ice thickness to Farinotti et al. (2019) (MAE = 4 m; Figure 4a), a minimal RGI6.0 area bias (Figure 3 and 4b), and very stable-state glacier conditions for > 500 years (Figure 3). This configuration also minimized errors against the Millan et al. (2022) dataset (MAE = 24 m) (Figure 4c). Thus, glacier reconstruction and projection are based on this parametrization option.

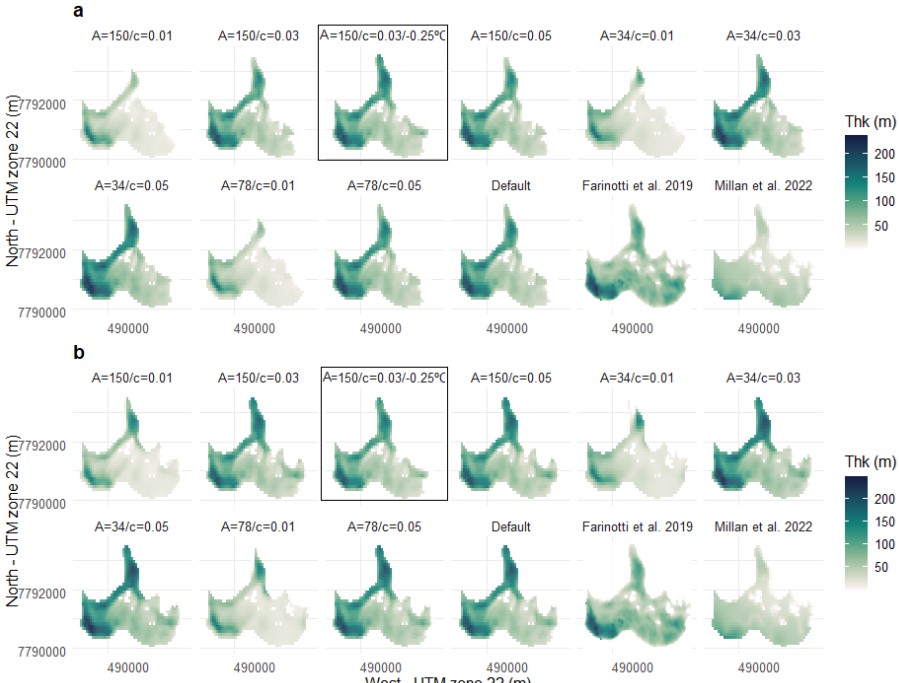

324

**Figure 5.** Average ice thickness data from Farinotti et al. (2019) and Millan et al. (2022),
along with examples of IGM configuration options using different A and c parameters for
the selected configuration (highlighted with a black square), are shown for a temperature
of 0ºC with respect to the baseline climate (a) and a temperature of -0.5ºC with respect to
the baseline climate (b). The ice thickness values of IGM shown are the result of steady-
state glacier conditions.

## 5.2 Late Holocene maximum glacier extension and paleoclimatic conditions

The temperature evolution from the LGM to 2000 CE, as reconstructed from GrIS ice
cores and Greenland margins (Buizert et al., 2018), is shown in Figure 6. The Camp
Century and Disko Bugt/Jakobshavn ice cores exhibit similar temperature trends
compared to the baseline climate period, although they display larger temperature
anomalies, with the warmest conditions recorded during the Holocene Warm Period
(HWP; ~ 9-5 ka ago) (up to 3ºC with respect to the baseline climate period). A long-term
cooling trend is detected for the Late Holocene, with moderate anomalies and high yearly
oscillations of around ±1 ºC between the Dark Ages Cold Period (~ 400 to 765 CE;
Helama et al., 2017) and the MWP for Disko Bugt/Jakobshavn (Figure 6). However,
colder temperatures are found in the Camp Century ice core. For both locations, the
coldest temperature anomalies of ca. -2ºC compared to the baseline climate are found
during the LIA.

344





345

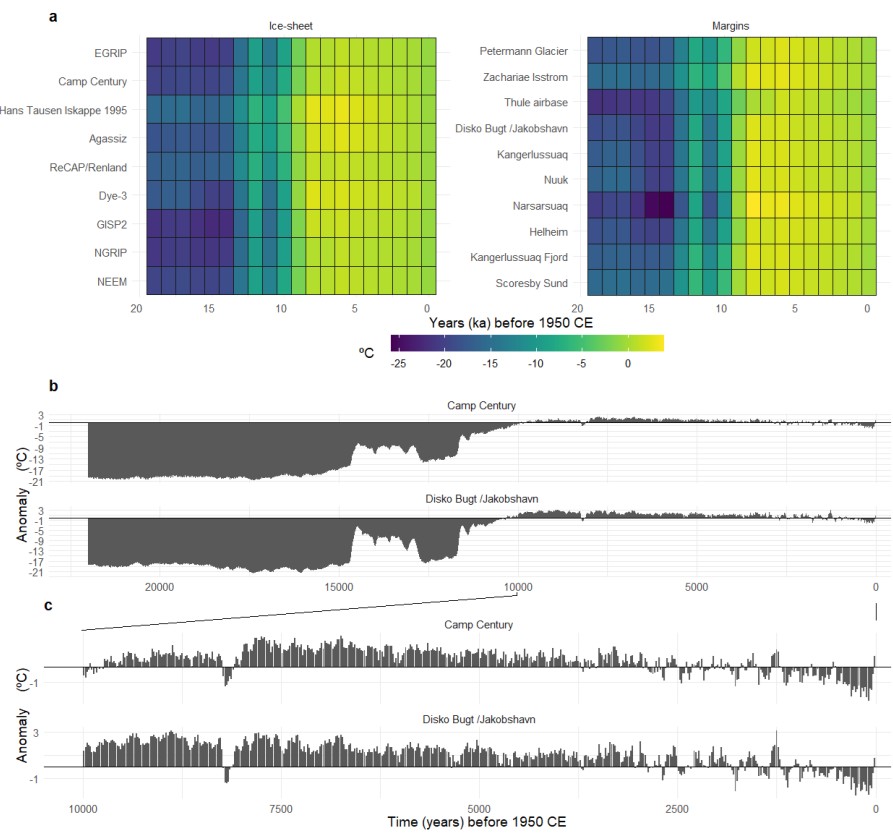

346

**Figure 6.** Air temperature anomalies from the LGM period to the present reconstructed from ice-core records of the GrIS and Greenland margins (a). Air temperature anomalies from the ice cores near to the glacier reconstructed in Central-Western Greenland (Camp Century and Disko Bugt/Jakobshavn) (b). The black squares highlighted in (a) correspond to the specific locations shown in (b). Air temperature anomalies since 10 ka to 1950 (c). Air temperature anomalies are calculated by the difference between the average annual air temperature from the baseline climate (1960-1990 period) and the annual air temperature for each location. Data were obtained from the reconstruction available from Buizert et al. (2018).

Future CMIP6 SSP2-4.5 and SSP5-8.5 anomalies with respect to the baseline climate are shown in Figure 7. The temporal evolution of temperature follows a similar warming rate for both scenarios until 2040, after which there is an acceleration of warming for SSP5-8.5. The increase in temperature relative to the baseline climate for SSP2-4.5 is 3.1ºC by 2050 and 4.2ºC by 2100, whereas for SSP5-8.5, is 3.4ºC by 2050 and 6.1ºC by 2100. Thus, CMIP6 (SSP2-4.5 and SSP5-8.5) anomalies with respect to the baseline climate are similar to HWP for 2050 but higher by a factor of three for SSP5-8.5 by 2100 (Figure 6).





Precipitation also shows an increase with respect to the baseline climate, which is more
pronounced for SSP5-8.5 towards the end of the 21st century. SSP2-4.5 precipitation
anomalies with respect to the baseline climate are +20% by 2050, increasing to +26% by
2100. For SSP5-8.5, precipitation increases by +20% by 2050 and +38% by 2100.
For the 2050-2060 period, summer temperatures are projected to range from 2ºC under
SSP2-4.5 to 3ºC under SSP5-8.5. For the 2090-2100 period, winter temperatures are
projected to range from 5ºC under SSP2-4.5 to 8ºC under SSP5-8.5 (Figure S1).
Regarding snowfall and for the 2050-2060 period, SSP2-4.5 and SSP5-8.5 projects
anomalies of 12% and 16 %, respectively (Figure S2 and S3). For the 2090-2100 period,
anomalies with respect to the baseline climate are 18 % and 22% for SSP2-4.5 and SSP5-
8.5, respectively. Other months show decreases in snowfall except for Spring, which
shows a 3% increase for both SSP5-8.5 and SSP2-4.5 scenarios for 2050-2060 and 2090-
2100 periods.

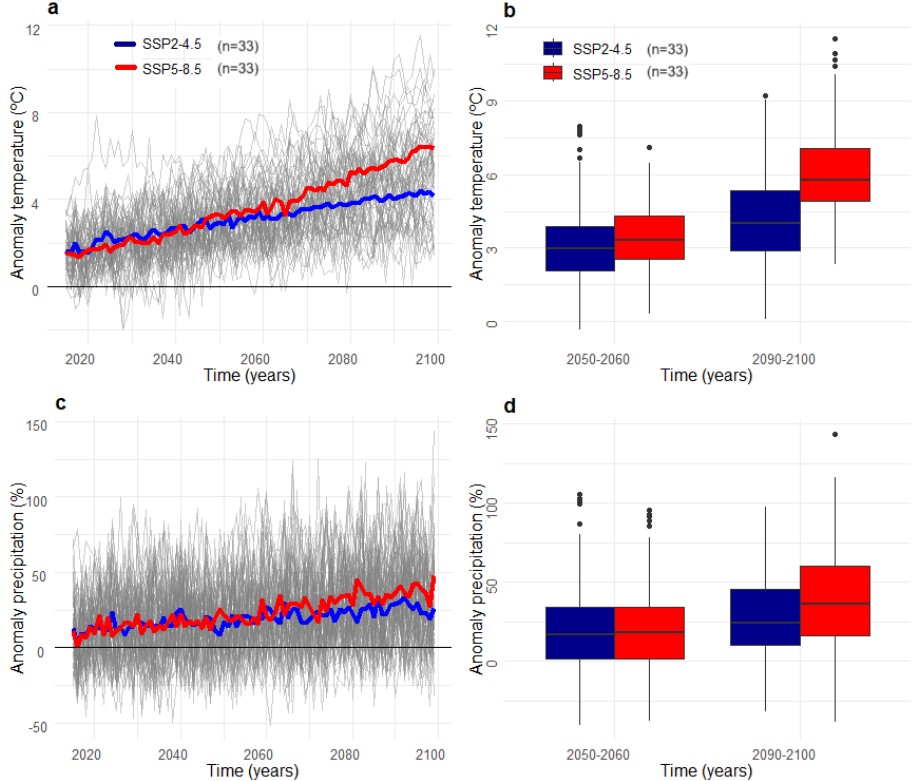


**Figure 7.** Temporal evolution of CMIP6 SSP2-4.5 and SSP5-8.5 temperature anomalies
with respect to the baseline climate period (a). Comparison of CMIP6 SSP2-4.5 and
SSP5-8.5 temperature anomalies with respect to the baseline climate period for 2050-
2060 and 2090-2100 temporal periods (b). Figure 7 (c) and (d) are the same as Figure 7



(a) and (b), respectively, but for precipitation. The dots of (b) and (d) represent the average of each CMIP6 model for the temporal period and climate variable.

We further assessed whether the temperature conditions reconstructed from ice-core data are consistent with CRE dates from moraine boulders and can accurately replicate the MIE of the Late Holocene. The IGM was spin-up and forced with the lowest error configuration. Subsequently, a sensitivity analysis of temperature and precipitation was conducted. The IGM was run after present-day steady-state conditions, with variations of temperature from 0 to -1 by steps of 0.25ºC. Precipitation was increased by 10%. We determined the temperature and precipitation conditions that allowed the MIE of the Late Holocene glacier extension, enabling its reconstruction (Figure 8).

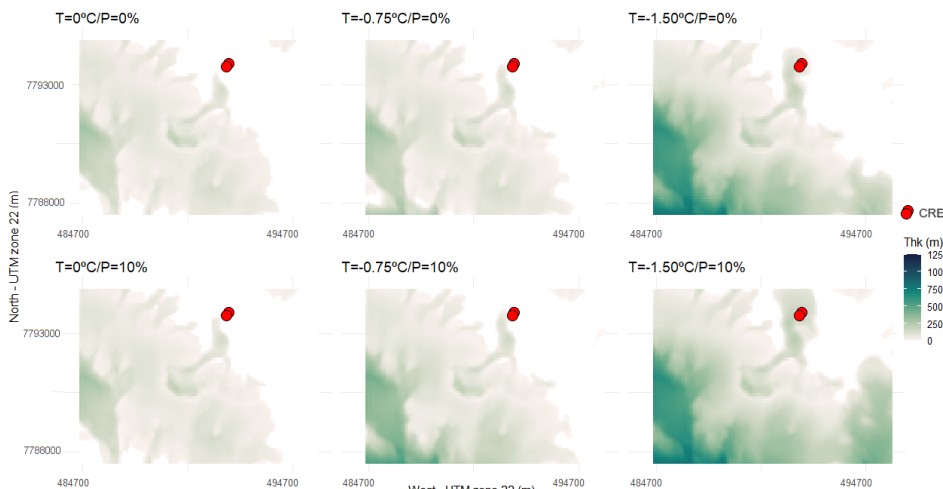

**Figure 8.** Location of the CRE samples (red dots) and average ice thickness (m) for various temperature (T) and precipitation (P) perturbations. The ice thickness values shown are the result of performing a spin-up model run, and subsequently a 1000-year model run for reconstructing the MIE of the Late Holocene.

The assessment of past temperature and precipitation anomalies relative to the baseline climate is conducted based on the distance between the glacier tongue and available CRE dates. This analysis indicates the temperature and precipitation conditions that facilitated glacier expansion during the MIE of the Late Holocene. Note that there may be a time gap between MIE of the Late Holocene the timing of maximum ice expansion and CRE ages and the since these ages indicate not the period of glacial growth but rather the period when moraine boulders stabilized after the formation of the moraine ridges formed by the glacier advances/stillstands. The minimum distance for all samples is reached when temperature is reduced by 0.5ºC and precipitation is increased by 10% with respect to the baseline climate. A reduction of 0.75ºC while maintaining precipitation unchanged resulted in glacier advances to the limit marked by the dated moraine boulders (Figure 8 and 9). These findings suggest that temperature anomalies leading to glacier extension up



to the MIE of the Late Holocene ranged at least from temperatures of -0.5ºC and
precipitation of +10% to temperatures of <= -0.75ºC and precipitation of 0% relative to
the baseline climate (Figure 9). However, a variation in precipitation of 10% is unlikely
according to paleoclimate reconstructions for the Late Holocene (Badgeley et al., 2020).
This suggests that a temperature decrease of at least 0.75ºC from the baseline climate,
with no changes in precipitation, is the most plausible climate scenario.

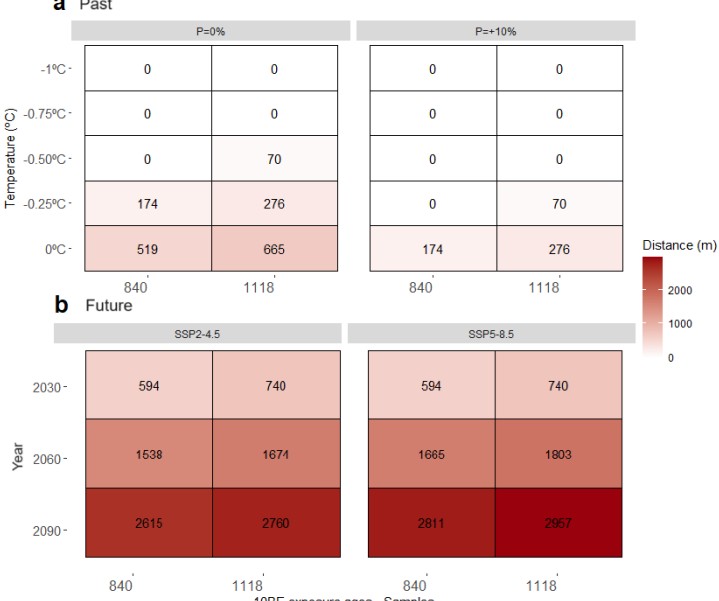

**Figure 9.** Differences in pixel distance (m) between the nearest modelled glacier
extension and the sample age location (x-axis) across various air temperature (y-axis) and
precipitation options (boxes) (a). Differences in pixel distance (m) between the nearest
modelled glacier extension and the sample age location (x-axis) and projected glacier
shrinkage for CMIP6 scenarios (boxes) and different years (y-axis).

The results suggest that with -0.75 ºC and no changes in precipitation anomalies with
respect to baseline climate, the glacier ice thickness has reduced 34% with respect to
surface covered by glaciers during the MIE of the Late Holocene (Figure 10).

**5.3 Future glacier changes**

Assuming the PDD parametrization of the 2000-2020 period calibrated with geodetic data
(Hugonnet et al., 2021), the future climate for 2060 leads to glacier tongue recession
ranging from 1674 m (SSP2-4.5) to 1903 m (SSP5-8.5) relative to the MIE of the Late
Holocene (Figures 9 to 11). By 2090, glacier reduction is projected to reach up to 2760 m
(SSP2-4.5) or 2957 m (SSP5-8.5). These results indicate that the projected increase in
precipitation (Figure 7) is insufficient to offset glacier shrinkage. The rate of ice loss from





the MIE of the Late Holocene to the present (34%) will double after 2070 (Figure 10a),
regardless of the CMIP6 scenario. The rate of ice loss will increase by 2080, reaching
anomalies of 72% (SSP2-4.5) and 78% (SSP5-8.5). By 2100 and under SSP5-8.5, the
reduction in ice thickness will reach a maximum ice loss of 95% relative to the MIE of
the Late Holocene (Figures 10b and 11).

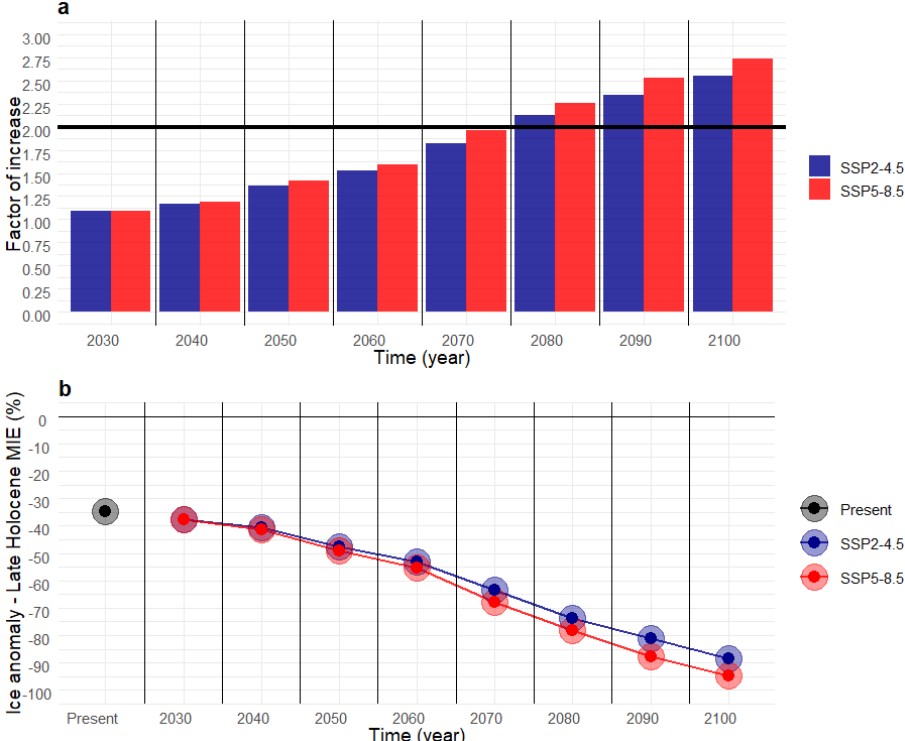


**Figure 10.** Factor of increase in ice loss under future climate change compared to ice loss
from the MIE of the Late Holocene to the present (a). Ice thickness anomalies for the
present and future CMIP6 SSP2-4.5 and SSP5-8.5 scenarios (b). Anomalies are calculated
by subtracting the accumulated yearly ice thickness for the MIE of the Late Holocene (i)
from the accumulated yearly ice thickness from the present-day (ii) and future ice-loss
changes (iii), dividing by (i), and multiplying by 100. The factor of increase under future
climate change is calculated by dividing future ice loss anomalies by the present-day ice
loss anomalies relative to the MIE of the Late Holocene.






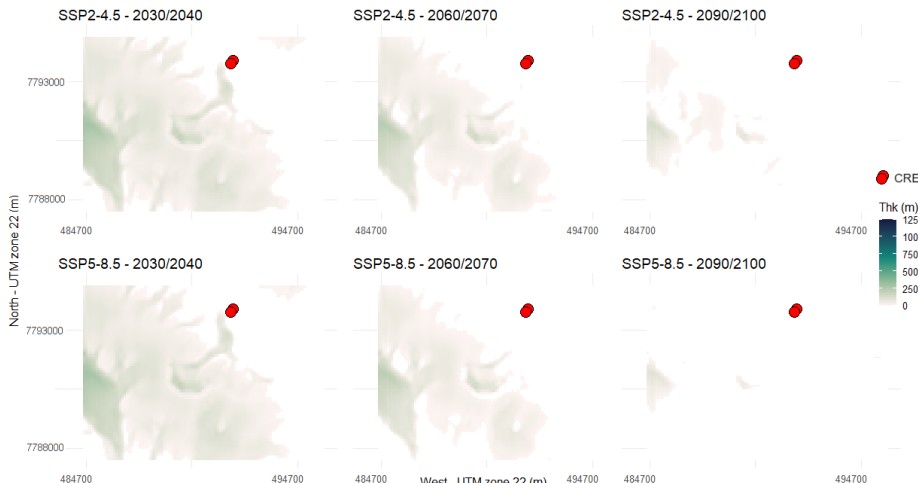

**Figure 11.** Location of the CRE samples (red dots) and average ice thickness (m) for future CMIP6 SSP2-4.5 and SSP5-8.5 scenarios and different temporal periods. The ice thickness values shown are the result of performing a spin-up model run reaching steady state conditions, and subsequently performing a model run with CMIP6 projections from present-day to 2100.

## 6. Discussion

### 6.1 Glacier modelling as a tool to understand paleoclimate conditions

The range of temperature decrease (0.75ºC to 1ºC) that obtained the best results in terms of reproducing glacier's MIE of Late Holocene area is consistent with past temperature anomalies in the Western and Southern Greenland found in previous works. Particularly, this range of temperature anomalies fall between estimates of ~1.5 ºC cooler temperature at 1850 CE with respect to 1990s (Dahl-Jensen et al., 1998). In Southern-Western Greenland, temperature estimates derived from geospatial reconstruction of ELAs that attributed historical MIE to the LIA, suggest temperatures ranging from around -0.4 to -0.9 ºC (Larocca et al., 2020). Employing a similar methodology, other studies have found temperature anomalies during the LIA to be of -1.1 ± 0.6 ºC, with no observed changes in precipitation (Brooks et al., 2022).

However, the MIE of the Late Holocene in Central-Western Greenland defined by the most recent moraine complexes suggest an earlier maximum glacier extent than in other areas in the Northern Hemisphere when the LIA glacier expansion was much more extensive (Young et al., 2015). In this sense, the temperature indicated by at Northern-Hemispheric scale reanalysis for the MWP is not consistent with MIE obtained from glacier moraines dated with CRE in Western Greenland (Jomelli et al., 2016; Biette et al., 2019). Indeed, Biette et al. (2019) modelled the outlet glacier of the Lyngmarksbræen ice cap (Disko Island) and tested its sensitivity to temperature and precipitation using a PDD approach guided by temperature anomalies from a lake sediment located 250 km south of



Disko Island (D'Andrea et al., 2011). They demonstrated that the MIE of the Late
Holocene during the late MWP (1200 ± 130 CE) occurred when temperatures ranged
from -1.3°C to -1.6°C, and precipitation changed by ± 10% (Biette et al., 2019).
Considering that the baseline climate period of our work is 1960 to 1990, and their
anomalies are considered to the end-20th century, our results are similar to these reported
temperature and precipitation values. These results are in line with a decrease in summer
temperature from -0.5º to -3ºC at around 250 km of Disko Island during the MWP
obtained from lacustrine (alkenone-based lake sediment) reconstructions, and a second
cold phase during the LIA(D'andrea et al., 2011). These cold conditions have been linked
to multi-decadal cold spells intense enough to cause a major advance of Baffin Bay during
the MWP (Young et al., 2015; Jomelli et al., 2016). These glacier advances were also
observed in other Northern Hemisphere glaciers and may be also enhanced by volcanic
eruptions (Solomina et al., 2016). The reconstructed glacier advances were probably
linked with a recurrent positive NAO at West Greenland and Baffin Bay during MWP
that lead to cool conditions (Young et al., 2015). However, other studies suggested that
the NAO was not predominantly positive during this period (Lasher and Axford, 2019).
There are also studies suggesting cold sea-surface temperatures observed during the
MWP (Sha et al., 2017), while others suggest warmer conditions during the MWP
compared to the LIA (Perner et al., 2012).
The MIE of the glaciers in the study area was reached during the MWP (1130 ± 40 and
925 ± 80 CE) (Young et al., 2015), has been suggested to be linked to increased snowfall
rates that counterbalanced the glacier ablation mass losses during the MWP, which were
slightly higher than those during the LIA (Osman et al., 2021). However, an increase of
precipitation of + 10 % with respect to the baseline climate period is not able to
counterbalance glacier recession under a change of <-0.5ºC with respect to the baseline
climate (Figure 9). The sensitivity analysis performed here reveals that the glaciers were
far isothermal conditions within 1960-1990 period and a small decadal variation of
temperature with respect to the baseline analysis to temperature and precipitation
performed in this work is consistent with previous works that suggest that around 90% of
variation and that glacier maximum extension dynamics are linked with summer
temperature (Miller et al., 2012; Young et al., 2015).

## 6.2 Central-Western Greenland ice-loss and comparison with other Greenland areas

The reconstructed MIE of the Late Holocene represents the phase with most recent
widespread glacier advances from the Nuussuaq Peninsula, and it occurred prior to the
LIA (Schweinsberg et al., 2017; 2019). The maximum glacier advance reconstructed in
this work for Central-Western Greenland is not consistent across Greenland. Northern
GrIS exhibits stability during the Late Holocene with advances at 2.8 ka and 1650 CE
(Reusche et al., 2018). In particular, North-Western glaciers length was similar from 5.8
ka until onset of LIA (Søndergaard et al., 2020). In the Bregne Ice Cap (East Greenland)
glacier length dating reveals a peak during the LIA (~ 0.74 ka; Levy et al., 2014).
However, in Renland Ice Cap (Eastern Greenland) glacier exceeded present limits at 3.3



ka and around 1 ka, which is similar to LIA glacier advance (Medford et al., 2021). In
Central-East Greenland, cold climate conditions occurred during LIA at Stauning Alper
with peaks of $0.78 \pm 0.31$ ka (Kelly et al., 2008), and at Istorvet ice cap, that reached its
maximum Holocene extent at $0.8 \pm 0.3$ ka (Lowell et al., 2013). This expansion observed
in Eastern Greenland corresponds with peak glacier extensions seen in Iceland, attributed
to LIA (Flowers et al., 2008). Different asymmetries between Greenland sectors are seen
historically as revealed by long-term GICs recession larger in West Greenland than in
East, which has been attributed to the positive oscillation of NAO since the LIA that led
to warmer conditions in West Greenland due to the West-East NAO dipole (Bjørk et al.,
524    2018).

In Central-Western Greenland, most of the studies focusing on Late Holocene glacial
history come from near Disko Island (Ingolfsson et al., 1990; Humlum, 1998; Yde and
Knudsen, 2007; Citterio et al., 2009; Jomelli et al., 2016). Here, in its Eastern fringe, the
ELA from the LIA is estimated at ca. $550 \pm 500$ m, contrasting with values of 200-300 m
attributed elsewhere in the island (Ingolfsson et al., 1990). In the Western section,
however, the ELA during the LIA was estimated to be at $450 \pm 420$ m (Humlum, 1998).
As in Nuussuaq Peninsula, the Holocene maximum extension in Disko Island is
evidenced by moraine systems exhibiting a fresh, partly unvegetated appearance, with of
prevalence of *Rhizocarpon geographicum* in these moraines (Humlum, 1987). This
absence of Holocene moraine systems beyond the LIA moraines indicates that the
advance of LIA represents the maximum extension of this glacier since the Late Holocene
(Humlum, 1999). This moraine evidence has been used to estimate the ELA (Brooks et
al., 2022; Carrivick et al., 2023). Particularly, using geospatial methods Carrivick et al.
(2023) attributed this trimline to the maximum extent of LIA and concluded that
Greenland GICs lost 499 Gt since end-LIA, corresponding to 1.38 mm sea level
equivalent. Similarly, in Southern-Western Greenland, 42 GICs lost 48% of their area
since the LIA with respect to 2019 (Brooks et al., 2022). These values are slightly higher
than the 34 % reduction from the MIE of the Late Holocene with respect to present-day
glacier reported in this work. The differences could be attributed to the local relief
configuration as well as to the north aspect of the reconstructed glacier area and
methodological variances. Additionally, while we are employing a glacier modelling
approach constrained by geological records of a specific age, previous studies have
estimated distances based on ELAs and geospatial methods that account for spatial
distances between present-day glaciers tongue and maximum historical moraines that
could be formed prior to the LIA. According to remote sensing data, in Disko Island GICs
inventory and monitoring from 1953 to 2005 indicates that the average recession during
this timeframe amounted to 11% of the glacier lengths recorded in 1953 (number of
glaciers, $n = 172$), and 38% of the distance between LIA moraines and glacier termini in
1953 ($n = 87$) (Yde and Knudsen, 2007). These values are lower than those observed at
Pjetursson Glacier (Disko Island), which has retreated since the LIA with a decrease in
total glacier area of around 40% by the end of the 20[th] century according to geospatial
methods (Bøcker, 1996). Using remote sensing data, LIA to 2001 glacier shrinkage in



Central-Western Greenland was estimated in a reduction of ~ 20% of the area (Citterio et
al., 2009).
Currently, the modeled glacier area and volume are out of balance with respect to the
temperature since 1990 to present (figure not shown), necessitating the simulation of
glaciers using temperature and precipitation data from the 1960-1990 period (Figure 2).
This indicates a committed ice loss regardless of future climate scenarios. Future
projections show a remarkable increase in temperature, reaching HWP anomalies by 2050
and tripling HWP anomalies by 2100 under SSP5-8.5. Our results indicate that glacier
mass loss by >2070 will double the ice loss from the MIE of the Late Holocene to the
present. Precipitation is projected to increase by 20% (2050; SSP2-4.5 and SSP5-8.5) up
to 38% (2100; SSP5-8.5) compared to the baseline climate but cannot counterbalance
glacier losses. The modeled GICs mass loss is expected to reach MIE of the Late
Holocene anomalies of 95% by 2100 under SSP5-8.5. The data presented in this work
suggests that future glacier ice loss will occur at unprecedented rates compared to the
period from the MIE of the Late Holocene to the present.
Currently, the modeled glacier area and volume are out of balance with respect to the
temperature since 1990 to present (figure not shown), necessitating the simulation of
glaciers using temperature and precipitation data from the 1960-1990 period (Figure 2).
This indicates a committed ice loss regardless of future climate scenarios. Future
projections show a remarkable increase in temperature, reaching HWP anomalies by 2050
and tripling HWP anomalies by 2100 under SSP5-8.5. Our results indicate that glacier
mass loss by >2070 will double the ice loss from the MIE of the Late Holocene to the
present. Precipitation is projected to increase by 20% (2050; SSP2-4.5 and SSP5-8.5) up
to 38% (2100; SSP5-8.5) compared to the baseline climate but cannot counterbalance
glacier losses. The modeled GICs mass loss is expected to reach MIE of the Late
Holocene anomalies of 95% by 2100 under SSP5-8.5. The data presented in this work
suggests that future glacier ice loss will occur at unprecedented rates compared to the
period from the MIE of the Late Holocene to the present.
According to CMIP6 projections for near-ice-free zones of Disko Island, this temperature
increase is explained by increases in long-wave radiation and slight variations or
decreases in short-wave radiation (Bonsoms et al., 2024). Future winter temperatures are
expected to remain below isothermal conditions, leading to more snowfall during winter
(i.e., +22% for SSP5-8.5 for the 2090-2100 period, relative to the baseline climate). The
increase of snowfall, however, cannot counterbalance glacier shrinkage, and a 10%
increase in precipitation has minimal impact on glacier area and thickness variability
(Figure 8). Snowpack projections for a near-ice-free region of Disko Island align with
these findings, indicating decreases in snow depth and snowfall fraction, along with
increases in snow ablation (Bonsoms et al., 2024a). For the GrIS, previous studies
projected a larger SMB decrease in ice sheet margins due to higher melting and lower
accumulation compared to the GrIS interior; pointing out that increases in snowfall are
insufficient to counterbalance the increased runoff (Fettweis et al., 2013). Yet, CMIP6



models are unable to capture the increase in anticyclonic events in Greenland since 1990s
(Delhasse et al., 2021), which have driven increased melting and extreme melting events
in the GrIS (Bonsoms et al., 2024).
Greenland GICs numerical modelling reconstructions are scarce in comparison with GrIS
numerical modelling works; including paleoclimate modelling (Huybrechts, 2002),
model parameters sensitivity studies (Cuzzone et al., 2019) or GrIS Holocene evolution
constrained with geological records (i.e., Simpson et al., 2009; Lecavalier et al., 2014,
Briner et al., 2020). GICs make a modest (11 %) contribution to total Greenland ice loss
but exhibit a fast response to warming (Khan et al., 2019). While we modeled the response
of glaciers in a Central-Western GIC area, future studies should compare these ice loss
rates with GrIS trends, which exhibit a slower response to warming (Ingolfsson et al.,
1990). The anticipated glacier retreat has important environmental implications,
including increased freshwater release into the North Atlantic and alterations in
atmospheric and circulation patterns (Yu and Zhong, 2018), which may impact the
Atlantic Meridional Overturning Circulation (Thornalley et al., 2018). Thus, Greenland
glacial retreat, snow melting, and permafrost thaw will amplify greenhouse gases release
and potentially trigger major consequences at global scale (Miner et al., 2022). Negative
mass balances will change geomorphological and permafrost patterns (Christiansen et al.,
2010) and ecosystem dynamics in ice-free zones, by modifying maritime (Saros et al.,
2019), and terrestrial phenological and fauna distribution (John Anderson et al., 2017).

### 6.3 Atmospheric forcing and numerical modelling considerations

This work is based on GSWP3 W5E5v2.0 climate dataset, which is based on ERA5
reanalysis data bias-adjusted over land (Lange et al., 2021). ERA5 incorporates
observations via a data-assimilation system combining observations, modelling, and
satellite data, and was previously validated in Greenland (Delhasse et al., 2020). ERA5
has been used to force state-of-the-art regional climate models, showing good agreement
with observations (Box et al., 2022). Our results are consistent with previous works that
provided a glacier reconstruction based on outputs of MAR forced with ERA5 and a PDD
model in Disko Island (Central-Western Greenland) (Biette et al., 2019). Results are
indeed similar to geo-spatial reconstructions in other Greenland sectors (i.e., Brooks et
al., 2022). The main conclusions of this work are consistent with paleo GrIS
reconstructions and projections in Central-Western GrIS (Briner et al. 2020).
A more sophisticated glacier modelling experiment will require data from coupling
regional circulation models, which account for changes in large-scale circulation.
However, glacier modelling driven by paleoclimate simulations has uncertainties and
large variability between models, as previous works in the study area have shown that
paleoclimate simulations cannot reconstruct Late Holocene glacier dynamics in the study
area (Jomelli et al., 2016; Biette et al., 2019). Paleoglacier modelling forced with
convection-permitting models is computationally demanding, relies on
parameterizations, and has limitations in simulating paleoclimate variables (Russo et al.,
2024). In this work a sensitivity analysis to precipitation and temperature is conducted to



reconstruct glacier MIE based on cosmogenic data, and therefore results are analyzed
based on anomalies with respect to a baseline climate (1960-1990), which is sufficiently
long to consider climate interannual variability and is marginally affected by climate
warming.
As with most paleo glacier models, IGM relies on a PDD approach, which is an
approximation that does not account for the Surface Energy Balance (SEB) driving
melting. However, the SEB components required for glacier modelling are uncertain for
the spatial and temporal scales analyzed in this study. PDD is based on a temperature
index model. Impurities on the ice (such as algae, dust, etc.) are not directly considered
but indirectly inferred by the melt rate factor. The IGM configuration for the calibration
and correction process of precipitation and temperature is based on OGGM v1.6.1
(Maussion et al., 2015; Schuster et al., 2023). This calibration corrects precipitation and
temperature to match geodetic mass balance at the glacier level (Hugonnet et al., 2021).
This product was selected due to the lack of long-term past and present in-situ mass
balance measurements in the study area. Errors of Hugonnet et al. (2021) product are
therefore influencing the glacier modelling results. The OGGM v1.6.1 calibration of bias
correction has been recently compared and cross-validated for glacier modelling of past
and future glacier projections, demonstrating reliable results (i.e., Aguayo et al., 2023;
Zekollari et al., 2024, and references therein).
IGM has been previously validated for modelling the present and projecting the future
evolution of glaciers, being successfully applied to the present and future scenarios of
alpine glaciers and providing reliable results (Cook et al., 2023); and references therein).
Here we have performed a IGM parameter tuning to accurately simulate present-day
glacier conditions. We cross validated results against two independent ice thickness
products (Farinotti et al., 2019; Millan et al., 2022) and RGI6.0 observations. Data shows
good agreement when compared to Farinotti et al. (2019) but lesser agreement against
Millan et al. (2022) (Figure 4). These differences could be attributed to the different
glacier methodologies: Farinotti et al. (2019) is based on an ensemble of five glacier
models founded on ice flow physics, whereas Millan et al. (2022) is based on glacier flow
mapping. Further research should analyze these differences. As most numerical modelling
experiments, past and future ice flow parameters are likely different from present-day
parameters due to unknown variables such as variations in basal conditions, bedrock
topography, and ice rheology. Consequently, IGM parametrization should be seen as a
simplification when applied to past and future conditions due to the difficulty of inferring
these parameters accurately.

**7. Conclusions**
This work analyzes the long-term dynamics of Central-Western GICs Greenland's and
their response to climate variability. We integrated ancillary data, ice thickness estimates



and geological records to increase the understanding of paleoclimate conditions in this
zone and contextualize present and future glacier loss within the Holocene.
The IGM underwent calibration and validation with various parametrization options of A
and *c* to accurately replicate glacier ice thickness and area. Following a long-term spin-
up simulation, the model converged to stable glacier conditions, matching available ice
thickness data and RGI6.0 area obtained from satellite observations and glacier
modelling. The optimal configuration reproduced available ice-thickness estimates,
representing an error of <10% of the total accumulated ice thickness for the modelled
area. Subsequently, the model was forced with an ensemble of temperature and
precipitation options, which were validated with CRE records, allowing to quantify
current glacier retreat since MIE of the Late Holocene. Further, IGM was forced with
CMIP6 projections towards 2100, allowing us to compare past and future recession within
a changing climate.
Results show that past glacier extensions during the MIE of the Late Holocene were
reached with temperature reductions that were likely to be between -0.75ºC to -1ºC with
respect to the baseline (1960-1990) climate period. Present-day reductions in glacier area
are 34% with respect to MIE of the Late Holocene. Results demonstrate the current
imbalance of Central-Western GICs and quantify how unprecedent are glacier shrinkage
within the Late Holocene. Future climate change will double the ice loss from Late
Holocene to present by > 2070. By 2100 and under SSP5-8.5, glacier mass is projected
to be reduced 95 % with respect to the MIE of the Late Holocene, with implications for
regional hydrology, ecosystems, and sea-level rise. The results provide a better
understanding of the response of Arctic peripheral glaciers and ice caps to climate change,
anticipating the formation of new landscapes, deglaciated areas, and lakes.

**Code and data availability**
IGM is an open-access model provided at https://github.com/jouvetg/igm (Jouvet, 2023).
Data of this work are available upon request to the first author (josepbonsoms5@ub.edu).
**Author contributions**
JB, MO and JILM conceptualized and designed the work. JB wrote the manuscript. JB,
MO and JILM edited the manuscript and contributed to the discussion of the results. JB
led the modelling of the work guided by GJ. GJ provided comments on the modelling
aspects of the manuscript. MO and JILM supervised the project and acquired funding.
**Competing interests**
The authors have not competing interests.
**Acknowledgements**
This manuscript falls within the research topics examined by the research group Antarctic,
Arctic and Alpine Environments (ANTALP; 2017-SGR-1102) funded by the Government
of Catalonia and MARGISNOW (PID2021-124220OB-100), from the Spanish Ministry
of Science, Innovation and Universities. Josep Bonsoms is supported by a pre-doctoral



FPI grant (PRE2021097046) funded by the Spanish Ministry of Science, Innovation and
Universities.

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
