# Peer review of "Tracing ice loss from the Late Holocene to the future in Eastern Nuussuag, 1 Central-Western Greenland 2 Josep Bonsoms 1\*, Marc Oliva 1, Juan Ignacio López-Moreno 2, Guillaume Jouvet 3 3 1\* Department of Geography, Universitat de Barcelona, Spain. 4 5 Email address: josepbonsom"

_EGUsphere, 2024_

## Referee Comment (RC1)

**Summary**

This paper presents numerical model simulations of a glacier in West Greenland. The authors aim to recreate the maximum position of the Holocene ice, in an attempt to compare ice loss rates since this period of time to contemporary and potential future loss rates. This is an interesting line of enquiry, which utilises the geomorphological/geochronological record and combines it with newly developed GPU accelerated ice flow modelling in a reasonably novel way. However, there are several major shortcomings of this work, outlined below, which I think need addressing before publication. I hope these comments improve the paper, as I like the overall approach, but I believe the following needs to be addressed for purposes of rigour and because you have the opportunity to have a good paper here.

**Major comments**

**Figures**

Firstly, and hopefully a quick fix, the figures are in general quite poorly designed and some of the figures are indecipherable. This makes it really difficult to follow the paper. The following needs to be addressed as a minimum:

- Fig 1 is ok, if a little crowded.
- Fig 2 is useful, but why are there green ticks and little symbols?
- Fig 3 and 4 ok, but it is never stated what A and $c$ are in the text, or how they work within the model.
- Fig 5 is hard to see differences between the plots – perhaps anomaly plots against both reference datasets would work better?
- Fig 6, text too small.
- Fig 8 is indecipherable – where is the glacier, is the whole page glaciated?
- Fig 10b, what do the circles represent? Should these be points or crosses? The circle represents some sort of uncertainty?
- Fig 11. I can't see anything on here.

Perhaps odd to start a review with the, but the poor quality of the figures really hampered my ability to judge this paper.

**Approach**

As said in the summary, the overall line of enquiry is an interesting one. But, this could substantially be improved upon with some alterations to the approach.

The speed of IGM allows you to run numerous ensemble simulations, yet your results are mostly based upon one calibrated simulation. This prevents you from defining any uncertainty in your simulations, which given there is also uncertainty in the observations, diminishes the rigour of your approach. My suggestion would be to base the results on numerous "acceptable" simulations, i.e. those that  fit within the uncertainty of the data. This is true for both the PDD factors and the ice flow factors. There is certainly large uncertainty within the PDD factors that would fit the geodetic mass balance observations. Multiple PDD factors could also lead to potentially multiple ice flow parameters that fit the data.

I get that the PDD calibration has to be conducted first in the experimental design, logistically making it difficult to combine the sampling of PDD and ice flow parameters, but ideally you would vary these together. Perhaps sequentially creating a distribution of acceptable PDD parameters then seeing which ice flow parameters fit the data based on that distribution is an acceptable way forward.

A and *c* are key to the above, yet their definition is never provided in the text. I assume A is part of the flow law, and c a sliding exponent/parameter, but the reader is left guessing. The equation that defines these both is necessary here.

The above would allow for a more Bayesian inference based approach to your results, which would give your study more statistical grounding.

You also have a static approach to climate downscaling. This limitation is sort of covered in the discussion, but given this source of uncertainty, it might be worth exploring a range of lapse rates for both temperature and precipitation.

The past climate also has uncertainty in the data that is not addressed or accounted for.

Finally, the assumption that modern-day parameters represent the past is acknowledged too late in the discussion. It's the final point. I think this needs to come in earlier as well.

*Overstating results from one glacier*

There are lots of GICs where the extent during different periods of time is well known (especially little ice age which is very clear in Greenland: https://agupubs.onlinelibrary.wiley.com/doi/full/10.1029/2023GL103950

It is unclear to me, and potentially to other readers, why you chose this one glacier. It may have behaved differently to others. I am not saying you should therefore model more than one glacier, but statements like "glacier mass loss will double" in the abstract are hyperbole when based on one glacier. Mass loss of one glacier is projected to double is all you can say based on your work.

The title also needs to reflect this.

*Line-by-line comments*

L9. "*projected* future trends" The whole framing of this question could do with a little work.

L10-11. The way this sentence is phrased, I struggled to see what the "gap" was. Please simplify.

L13. The way this is written, I assumed you did the cosmogenic dating in this paper. Please clarify that this is from previous work.

L35. The glaciers aren't accelerating the mass loss themselves, they are displaying accelerated mass loss.

L73. I didn't understand this line. Please rephrase.

L189. It's not clear what the "shop module" is.

L235. This is where A and c need to be defined, with equations so we can understand their effect.

L237. What size is the ensemble? How were parameters sampled?

L256. I got confused here with the terminology. Is this an ensemble in the way it is commonly used (e.g. 100s-1000s of simulations with randomly perturbed parameters?) or a sensitivity test (systematically changing one variable?). You can do sensitivity analysis on an ensemble. Ensure your terminology is correct throughout.

Results and discussion: In my opinion, the above major flaws need to be considered regarding the approach. Which will likely require a major rewrite of this section.

L430-431. Just one example where you need to be clear that this is just one glacier. Please be consistent throughout.

L500. What are far isothermal conditions?

The paragraphs starting on line 559 and Line 572 are repeats. This repeated paragraph is problematic. Why is a figure not shown? Throughout there is a mix of statements pertaining to one glacier and glaciers in general.

L635. This statement on computational demands, parameters and limitations is a bit strange, your approach also has many of these limitations. I don't think a point is conveyed in this paragraph. I would consider removing or rewriting.

L676. GIC**s** is not correct here. Just one.

Conclusions need a rewrite in light of the above.

---

## Author Response (AR1)

**Editor:**
Dear authors,

Many thanks for your extensive work following both reviewer's comments. I believe that the results are more robust now. My only wish would be, if possible, to run just two simulations using a precipitation lapse rate of 50/100 and 70/100. This is because it is important to test the sensitivity of those projections and paleo reconstructions to precipitation, as it substantially impact on the SMB and thus on the glacier dynamics. Previous paleo-studies, using 3D ice sheet models with PDD for Greenland paleo reconstructions (e.g. Quiquet et al. and others) used lapse rates for precipitation between 50/100 and 70/100. As you stated that you prescribed the ad-hoc value of 35/100, I think it is fair to test a couple of other values. The rest of the manuscript is fine to me.

**Authors:**

Thank you very much for your feedback and for handling the manuscript.

I have carefully addressed your request by providing a detailed explanation of the SMB estimation and conducting an additional uncertainty analysis of the calibration process. This includes new model runs and an additional figure in the supplementary materials.

Importantly, and regarding the referenced work for the GrIS, different lapse rates were applied without correcting for biases, as their primary objective was to evaluate how these factors impact the ice sheet. In contrast, the model, calibration, and parameterization used in this work follow a different approach, which does not rely on arbitrary values. Instead, it incorporates corrections and calibrations based on the most recent and available mass balance products. The calibration is a state-of-the-art process introduced by Marzeion et al. (2012), implemented in OGGM (Maussion et al., 2019), and applied in IGM as well. This SMB estimation has been successfully applied in numerous studies since its introduction (Marzeion et al., 2012; Maussion et al., 2019), and it has been validated both prior to and within our work, through three independent sources (Figures 2 to 5), providing reliable results.

We have modified the methods description and, additionally, performed simulations using different lapse rates to account for potentially different values, including a low value (0.55°C/m), a mid-range lapse rate (0.65°C/m), and a high-end lapse rate (0.75°C/m). Furthermore, we have clarified the extrapolation and bias correction for both temperature and precipitation (see below).

These new model runs and the corresponding results have been added to the manuscript, along with figures in the supplementary material. This modification results in a 3% difference in the ice-thickness anomalies and introduces a third uncertainty analysis, complementing the uncertainty analysis of the ice flow configuration and melt rates.

All the best,

Josep Bonsoms on behalf of the co-authors.

**Changes:**

**4.1 Instructed Glacier Model (IGM)**

[revised manuscript text omitted]

**Response to the review of *"Projected climate change will double the Late Holocene maximum to present ice loss in Eastern Nuussuaq, Central-Western Greenland by 2070,"* submitted to *The Cryosphere*.**

Reviewer 1

Summary

This paper presents numerical model simulations of a glacier in West Greenland. The authors aim to recreate the maximum position of the Holocene ice, in an attempt to compare ice loss rates since this period of time to contemporary and potential future loss rates. This is an interesting line of enquiry, which utilises the geomorphological/geochronological record and combines it with newly developed GPU accelerated ice flow modelling in a reasonably novel way. However, there are several major shortcomings of this work, outlined below, which I think need addressing before publication. I hope these comments improve the paper, as I like the overall approach, but I believe the following needs to be addressed for purposes of rigour and because you have the opportunity to have a good paper here.

Authors: We would like to express our sincere gratitude for your review.

We have carefully revised the manuscript, making the necessary corrections based on his/her feedback. The results have been adjusted accordingly, and both the text and figures have been updated.

Please find a detailed response to your comments below. The reviewer's comments are shown in blue, while the author's responses are shown in black.

Major comments

Figures Firstly, and hopefully a quick fix, the figures are in general quite poorly designed and some of the figures are indecipherable. This makes it really difficult to follow the paper. The following needs to be addressed as a minimum:

Fig 1 is ok, if a little crowded.

Ok.

Fig 2 is useful, but why are there green ticks and little symbols?

These symbols aim to enhance the interpretability of Fig. 2, ensuring that readers can quickly grasp key validation aspects and the steps followed. We have added the following to the description: "The symbols next to the graph are included to make it easier to visually associate elements and enhance interpretability within the figure".

Fig 3 and 4 ok, but it is never stated what A and c are in the text, or how they work within the model.

Corrected. We have now incorporated the basic equations and provided a more detailed description of the A and c parameters. Please find below a comprehensive response to this aspect.

Fig 5 is hard to see differences between the plots – perhaps anomaly plots against both reference datasets would work better?

In this case, we believe it is better to show absolute values. Creating anomalies against the two reference datasets would result in 40 maps and 8 rows, making the interpretation of the map difficult.

Fig 6, text too small.

Done. We have corrected the text by increasing the text font size.

Fig 8 is indecipherable – where is the glacier, is the whole page glaciated?

We have clarified this point in the methodology section and added the following: "To accurately reconstruct the glaciated area, it is essential to model the region beyond the glacier using cosmogenic data. The IGM is applied to a region of interest in Eastern Nuussuaq, which includes 25 glaciers from the RGI6.0 database and covers a total area of 154 km²."
In short, to accurately reconstruct the glacier system, it was necessary to model the entire glaciated area, as modeling individual glaciers would introduce artifacts.

Fig 10b, what do the circles represent? Should these be points or crosses? The circle represents some sort of uncertainty?

This figure has been modified according to your suggestion (below).

We modeled the past and future simulations using calibrated, low, and high-end melt rates of the PDD. The error bars in the current figure represent confidence intervals.

We have added: "…the column bars represent the mean ice thickness anomalies, while the error bars indicate the standard deviation of the anomalies, reflecting the variability associated with the different melt rate factors."

Fig 11. I can't see anything on here.

We have clarified this point in the methodology section and added the following: "To accurately reconstruct the glaciated area, it is essential to model the region beyond the glacier using cosmogenic data. The IGM is applied to a region of interest in Eastern Nuussuaq, which includes 25 glaciers from the RGI6.0 database and covers a total area of 154 km²."

Perhaps odd to start a review with the, but the poor quality of the figures really hampered my ability to judge this paper.

Approach

As said in the summary, the overall line of enquiry is an interesting one. But, this could substantially be improved upon with some alterations to the approach. The speed of IGM allows you to run numerous ensemble simulations, yet your results are mostly based upon one calibrated simulation. This prevents you from defining any uncertainty in your simulations, which given there is also uncertainty in the observations, diminishes the rigour of your approach. My suggestion would be to base the results on numerous "acceptable" simulations, i.e. those that fit within the uncertainty of the data. This is true for both the PDD factors and the ice flow factors. There is certainly large uncertainty within the PDD factors that would fit the geodetic mass balance observations. Multiple PDD factors could also lead to potentially multiple ice flow parameters that fit the data. I get that the PDD calibration has to be conducted first in the experimental design, logistically making it difficult to combine the sampling of PDD and ice flow parameters, but ideally you would vary these together. Perhaps sequentially creating a distribution of acceptable PDD parameters then seeing which ice flow parameters fit the data based on that distribution is an acceptable way forward.

We have revised the manuscript based on the reviewer's suggestions and re-ran the simulations. Eight new figures have been added to the supplementary materials, and the existing ones in the main manuscript have been updated.

Regarding the PDD calibration, we used a calibrated melt rate factor, following the calibration approach against mass-balance data from Hugonnet et al. (2021). We incorporated both maximum and minimum melt rate factors. For glacier reconstruction, the calibrated melt rate represents the maximum, as it aligns with the last 20 years of dh/dt data. A low-end value (3) is introduced to represent the minimum melt rate. For future projections, the calibrated melt rate factor serves as the minimum, while a high-end factor (9) is also included. Adjusting the melt rate factors required recalibrating the model and introducing different temperature variations, as shown in Figures S1 and S2. These changes provide a confidence interval that defines the limits of both past reconstructions and future projections.

Additionally, we analyzed the IGM parameterization of ice dynamics and lapse rates (see first page of the review), and its influence on ice-thickness anomalies. We compared the IGM default configuration, which is valid for reproducing the present-day glaciated area, with the calibrated A and c parameters. The differences in ice-thickness anomalies were minimal (less than 4%) while maintaining the same temperature offsets (Figures S6 and S7). By incorporating these changes, we provide a range of possible outputs, ensuring consistency with the past, present, and future evolution of the glaciated area.

In the current version of the manuscript, we have added a more detailed description of the method.:

[revised manuscript text omitted]

A and c are key to the above, yet their definition is never provided in the text. I assume A is part of the flow law, and c a sliding exponent/parameter, but the reader is left guessing. The equation that defines these both is necessary here.

Indeed, A and c are key parameters of the IGM, and we have added the main formulation description of these parameters. For a more comprehensive explanation, we have included references where the reader can find an accurate description.

We have added:

"The ice flow is modelled using a CNN model that is trained to satisfy high-order ice flow equations. The strength of the ice flow is modeled through the rate factor (A) that controls the ice viscosity in Glen's flow law (Glen, 1955), expressed as:

$$\dot{D} = A\tau^n,$$

Where $\dot{D}$ and $\tau$ are the strain rate and deviatoric stress tensors, respectively and n is Glen's exponent, 3 (Glen, 1955). The basal sliding is modeled using the nonlinear sliding law of Weertman (Weertman, 1957), expressed as:

$$u_b = c\,\tau_b^{1/m},$$

Where $u_b$ is the sliding velocity, c is the basal sliding, $\tau_b$ is the basal shear stress, and $m$ is a constant of 1/3. The parameters A and c are parametrized (c.f. section 4.2) to reproduce available ice thickness datasets (Farinotti et al., 2019; Millan et al., 2022).

The explanation of why these parameters are changed can be found in Section 4.2:

"….The IGM is calibrated to simulate the RGI6.0 area and ice thickness using available datasets (Farinotti et al., 2019; Millan et al., 2022). The IGM parametrization was performed by conducting a sensitivity analysis to spin-up temperature and ice-flow dynamics, adjusting the parameters A and c. These parameters were selected to optimize the IGM and accurately simulate various ice conditions, basal sliding conditions, and subglacial hydrology. A set of parameter options (*n* = 36) was tested over a 1000-year model run to achieve long-term (>500 years) glacier area steady-state conditions and reproduce the RGI6.0 area and ice thickness from the available datasets (Farinotti et al., 2019; Millan et al., 2022). The calibration parameter options include different temperature perturbations. For a calibrated melt rate factor (see Section 4.1), temperature perturbations of -0.75ºC, -0.5ºC, 0ºC, and +0.25ºC are applied relative to

the baseline climate (1960–1990). For a low-end melt rate factor (3), temperature perturbations range from 0.75ºC to 1.25ºC in increments of 0.25ºC. For a high-end melt rate factor (9), temperature perturbations range from -1.75ºC to -1.25ºC in increments of 0.25ºC. The range of temperature perturbations was determined through trial and error, which showed that values outside this range produced higher discrepancies compared to the available datasets used for validation (Figures 3, 5, S1 and S2). Regarding ice-flow dynamics, a sensitivity analysis was performed on IGM parametrization to simulate cold, temperate, and soft ice conditions by changing A from 34 MPa$^{-3}$ a$^{-1}$, 78 MPa$^{-3}$ a$^{-1}$ (IGM default value) to 150 MPa$^{-3}$ a$^{-1}$. Basal sliding conditions are parametrized by changing c from 0.01 km MPa$^{-3}$ a$^{-1}$, 0.03 km MPa$^{-3}$ a$^{-1}$ (IGM default value), and 0.05 km MPa$^{-3}$ a$^{-1}$. The IGM parameterization is shown in Figures 3 to 5. An analysis of the influence of the IGM calibrated ice-dynamics options and the default configuration is also performed. All other parameters were kept at their default IGM configuration."

The above would allow for a more Bayesian inference based approach to your results, which would give your study more statistical grounding. You also have a static approach to climate downscaling. This limitation is sort of covered in the discussion, but given this source of uncertainty, it might be worth exploring a range of lapse rates for both temperature and precipitation. The past climate also has uncertainty in the data that is not addressed or accounted for. Finally, the assumption that modern-day parameters represent the past is acknowledged too late in the discussion. It's the final point. I think this needs to come in earlier as well.

Due to the lack of observational data in the region, we relied on the best-performing reanalysis product based on corrected ERA5 data, which assimilates the nearest meteorological observations and has been validated in Greenland (references provided). Temperature lapse rates were assumed to be similar to those reported in the literature (Hanna et al., 2005; Erokhin et al., 2017). An uncertainty analysis to the temperature lapse-rate is performed (see first page of this response, Results section and Figure S8).

We added a clarification in the methodology section: "The current lapse rate aligns with the annual lapse rates reported in the literature, such as 0.6ºC/100 m at low elevations of the GrIS (Hanna et al., 2005) and Disko Island (Humlum, 1998). This is similar to the values during the pre-industrial period and early Holocene (0.7ºC/100 m) (Erokhin et al., 2017)."

Overstating results from one glacier

There are lots of GICs where the extent during different periods of time is well known (especially little ice age which is very clear in Greenland: https://agupubs.onlinelibrary.wiley.com/doi/full/10.1029/2023GL103950 It is unclear to me, and potentially to other readers, why you chose this one glacier. It may have

behaved differently to others. I am not saying you should therefore model more than one glacier, but statements like "glacier mass loss will double" in the abstract are hyperbole when based on one glacier. Mass loss of one glacier is projected to double is all you can say based on your work.  The title also needs to reflect this.

**Title**

Changed to "Tracing Ice Loss from the Late Holocene to the Future in Eastern Nuussuaq, Central-Western Greenland" for clarity and conciseness. Terminology has been updated to refer to the "glaciated area", where applicable.

**Region of interest**

To accurately reconstruct the glaciated area, it is essential to model the region beyond the glacier using cosmogenic data. The IGM is applied to a region of interest in Eastern Nuussuaq, which includes 25 glaciers from the RGI6.0 database and covers a total area of 154 km²." The selected zone for reconstruction was chosen due to the availability of GIC cosmogenic surface dating over moraine boulders in the region. The reconstructed area spans several RGI6.0 glaciers and is now specified in the text, covering 154.1 km² (WGS 84 / UTM zone 22N).

**Referenced Study**

The referenced study addresses a similar objective but employs a different approach. It is based on a GIS tool for Equilibrium Line Altitude (ELA) estimation, calibrated with four dated moraines in Greenland corresponding to the Little Ice Age (LIA). However, in our study, we perform a different analysis to reconstruct the glaciated area using a physics-based numerical model informed by a CNN emulator. The calibration and validation of our model differ from the approach in the referenced work. Late-Holocene moraines are typically recognized as unvegetated trimlines in Greenland, but surface dating is required to estimate their ages, as the Late Holocene MIE varies across Greenland. According to values from independent sources, the Late Holocene ice extent in Central-Western Greenland is from before (e.g., Young et al., 2015; Jomelli et al., 2016; Schweinsberg et al., 2019).

"….The recent evolution of the GICs has been reconstructed using historical aerial images and satellite records (Leclerq et al., 2012; Yde and Knudsen, 2007; Citterio et al., 2009; Bjørk et al., 2018; Larocca et al., 2023). Geospatial techniques, such as the inference of the Equilibrium Line Altitude (ELA), have also been utilized (Brooks et al., 2022; Carrivick et al., 2023). However, aerial and satellite images provide temporal data over centuries and decades and geospatial methods neglect ice-flow physics and do not account for glacier dynamics. Based on the distribution of moraines and unvegetated trimlines in Central-Western Greenland, some authors suggested that the Late Holocene maximum glacier extent occurred around the LIA (Humlum, 1999). However, cosmic ray exposure (CRE) dating of erosive and depositional glacial records indicates that the maximum ice extent (MIE) of the Late Holocene did not occur during the LIA in many areas in Western Greenland but during the Medieval Warm Period (MWP; 950 to 1250 CE) (Young et al., 2015; Jomelli et al., 2016; Schweinsberg et al., 2019).

Line-by-line comments

L9. "projected future trends" The whole framing of this question could do with a little work.

Changed to "the extent to which projected future trends of GICs are unprecedented within the Holocene".

L10-11. The way this sentence is phrased, I struggled to see what the "gap" was. Please simplify.

Changed to: "This study bridges the gap between the maximum ice extent (MIE) of the Late Holocene, present and future glacier evolution until 2100 in Eastern Nuussuaq Peninsula (Central-Western Greenland)".

L13. The way this is written, I assumed you did the cosmogenic dating in this paper. Please clarify that this is from previous work.

Changed to "…The model is employed to reconstruct the Eastern Nuussuaq Peninsula GICs to align with the MIE of the Late Holocene, which occurred during the Late Medieval Warm Period (1130 ± 40 and 925 ± 80 CE), based on moraine boulder surface exposure dating from previous studies."

L35. The glaciers aren't accelerating the mass loss themselves, they are displaying accelerated mass loss.

Changed to "…glaciers are displaying accelerated mass loss (Hugonnet et al., 2021)"

L73. I didn't understand this line. Please rephrase.

Changed to: "….Compared to studies near the GrIS (e.g., Cuzzone et al., 2019; Briner et al., 2020), there is limited evidence from physically-based models regarding the GIC recession during the Holocene."

L189. It's not clear what the "shop module" is.

Changed to "module". IGM has a module that allows to download data from OGGM.

L235. This is where A and c need to be defined, with equations so we can understand their effect.

Dito: Changed. We have added this suggestion.

L237. What size is the ensemble? How were parameters sampled?

We have modified the word "ensemble" to avoid confusions and also modified he calibration process, where explains this poin (Section 4.2):

"...The IGM is calibrated to simulate the RGI6.0 area and ice thickness using available datasets (Farinotti et al., 2019; Millan et al., 2022). The IGM parametrization was performed by conducting a sensitivity analysis to spin-up temperature and ice-flow dynamics, adjusting the parameters A and c. These parameters were selected to optimize the IGM and accurately simulate various ice conditions, basal sliding conditions, and subglacial hydrology. A set of parameter options ($n$ = 36) was tested over a 1000-year model run to achieve long-term (>500 years) glacier area steady-state conditions and reproduce the RGI6.0 area and ice thickness from the available datasets (Farinotti et al., 2019; Millan et al., 2022). The calibration parameter options include different temperature perturbations. For a calibrated melt rate factor (see Section 4.1), temperature perturbations of -0.75ºC, -0.5ºC, 0ºC, and +0.25ºC are applied relative to

the baseline climate (1960–1990). For a low-end melt rate factor (3), temperature perturbations range from 0.75ºC to 1.25ºC in increments of 0.25ºC. For a high-end melt rate factor (9), temperature perturbations range from -1.75ºC to -1.25ºC in increments of 0.25ºC. The range of temperature perturbations was determined through trial and error, which showed that values outside this range produced higher discrepancies compared to the available datasets used for validation (Figures 3, 5, S1 and S2). Regarding ice-flow dynamics, a sensitivity analysis was performed on IGM parametrization to simulate cold, temperate, and soft ice conditions by changing A from 34 MPa$^{-3}$ a$^{-1}$, 78 MPa$^{-3}$ a$^{-1}$ (IGM default value) to 150 MPa$^{-3}$ a$^{-1}$. Basal sliding conditions are parametrized by changing c from 0.01 km MPa$^{-3}$ a$^{-1}$, 0.03 km MPa$^{-3}$ a$^{-1}$ (IGM default value), and 0.05 km MPa$^{-3}$ a$^{-1}$…."

L256. I got confused here with the terminology. Is this an ensemble in the way it is commonly used (e.g. 100s-1000s of simulations with randomly perturbed parameters?) or a sensitivity test (systematically changing one variable?). You can do sensitivity analysis on an ensemble. Ensure your terminology is correct throughout. Results and discussion: In my opinion, the above major flaws need to be considered regarding the approach. Which will likely require a major rewrite of this section.

We have modified the term "ensemble" to avoid confusion and have also clarified the calibration process. This is explained in detail in Section 4.2.

L430-431. Just one example where you need to be clear that this is just one glacier. Please be consistent throughout.

We modeled a glaciated area that includes 25 glaciers from the RGI6.0 database, covering a total of 154 km². To accurately reconstruct the ice-cap, it was necessary to model the entire glaciated system, as modeling the individual glacier would introduce artifacts. As the modeled area spans several glaciers in the RGI6.0 inventory, we have revised the text throughout to refer to the "glaciated area" rather than "glacier(s)."

L500. What are far isothermal conditions?

The phrase has been updated to: "…not in an isothermal state during the 1960–1990 period" to avoid any confusion.

This repeated paragraph is problematic. Why is a figure not shown? Throughout there is a mix of statements pertaining to one glacier and glaciers in general.

We opted not to include one figure, as we believe it would extend the analysis beyond the scope of this study. Our focus was not on analyzing the committed glacier changes in the region given the current climate conditions.

We have revised the text throughout to refer to the "glaciated area" rather than "glacier(s)."

Line 559 and Line 572 are repeats

Thank you for identifying this error. The repeated paragraph has been deleted.

L635. This statement on computational demands, parameters and limitations is a bit strange, your approach also has many of these limitations. I don't think a point is conveyed in this paragraph. I would consider removing or rewriting.

Done. The statement has been deleted.

L676. GICs is not correct here. Just one. Conclusions need a rewrite in light of the above.

We have revised the text throughout to refer to the "glaciated area" rather than "glacier(s)."

Conclusions are rewritten considering the comments of Reviewer 1 and 2:

"This study provides a long-term perspective on the dynamics of Eastern Nuussuaq, Central-Western Greenland's GICs in response to climate change. By integrating geological records, ice thickness estimates, and climate model projections, we contextualize present and future glacier loss within the Late Holocene.

The IGM was calibrated and validated using various parameterizations to accurately simulate glacier ice thickness and area. After a long-term spin-up simulation, the model stabilized, closely matching available ice thickness data and satellite observations from RGI6.0. The optimal configuration reproduced ice-thickness estimates with an error of less than 10% of the total accumulated ice thickness for the modelled area. Subsequently, the model was forced with an different temperature and precipitation scenarios, validated with CRE records, enabling the quantification of glacier retreat since the MIE of the Late Holocene. For future projections, IGM was driven by CMIP6 climate scenarios (SSP2-4.5 and SSP5-8.5), providing a comparative framework for past and future glacier recession in a changing climate. The main conclusions of this study are as follows:

- The MIE of the Late Holocene was reached when temperatures were 0.75°C to 1°C lower than the baseline climate period (1960-1990) under a calibrated melt rate factor.

- Currently, glaciated area ice thickness has retreated by 15% (low-end melt rate) to 20% (calibrated melt rate) compared to the MIE of the Late Holocene.

- Glacier mass loss is projected to occur at an unprecedented rate within the Late Holocene. Future simulations for 2070-2080 indicate a retreat more than double (-56±6%) compared to the ice loss from the MIE of the Late Holocene to the present.

- The glaciated area is expected to disappear towards 2090-2100.

Results confirm the ongoing imbalance of Eastern Nuussuaq, Central-Western Greenland GICs and highlight the unprecedented nature of current glacier shrinkage within the Late Holocene. Projections suggest that climate change will accelerate ice loss beyond historical trends, transforming Arctic landscapes, increasing deglaciated areas, and promoting the formation of new lakes. These findings enhance our understanding of Arctic peripheral glacier responses to anthropogenic climate change, with broad implications for hydrological and ecological systems."

**Response to the review of *"Projected climate change will double the Late Holocene maximum to present ice loss in Eastern Nuussuaq, Central-Western Greenland by 2070,"* submitted to *The Cryosphere*.**

I read with attention and interest the article by Bonsoms et al "Projected climate change will double the late Holocene maximum...", and found it well written and organized, full of results relevant not only to cryosphere/paleoclimate specialists, but also to those dealing with the impacts of climate warming on the Arctic ecosystem in general.

My opinion is positive and I have not found any formal or substantial weaknesses. It is a modelling paper, and therefore it is not easy to read for those who are not into this subject. The data are many, the text constantly refers to acronyms, and rightly some methodological aspects are not discussed in depth. I believe that this is the only way to write papers of this type, and in any case the results in terms of glacial extension and paleoclimate are clearly expressed and usable in other scientific contexts. The use of chronological data (CRE ages) relating to glacial landforms is a point of merit, which makes the modeling even more robust.

Authors: We want to express our sincere gratitude for your review.

Please find a detailed response to your comments below.

The reviewer's comments are shown in blue, while the author's responses are shown in black

The results are discussed in light of other data available for the same area and for the Arctic region in general, and the conclusions are concise. The authors may consider adding a short list of the main results at the beginning, to help the reader to recap before reading the final comments.

The abstract has been modified to improve readability and highlight the main conclusions for a more general audience, not specifically focused on modeling. The conclusions are modified according to the suggestion, adding a short list of the main results at the beginning:

"This study provides a long-term perspective on the dynamics of Eastern Nuussuaq, Central-Western Greenland's GICs in response to climate change. By integrating geological records, ice thickness estimates, and climate model projections, we contextualize present and future glacier loss within the Late Holocene.

The IGM was calibrated and validated using various parameterizations to accurately simulate glacier ice thickness and area. After a long-term spin-up simulation, the model stabilized, closely matching available ice thickness data and satellite observations from RGI6.0. The optimal configuration reproduced ice-thickness estimates with an error of less than 10% of the total accumulated ice thickness for the modelled area. Subsequently, the model was forced with an different temperature and precipitation scenarios, validated with CRE records, enabling the quantification of glacier retreat since the MIE of the Late Holocene. For future projections, IGM was driven by CMIP6 climate scenarios (SSP2-4.5 and SSP5-8.5), providing a comparative framework for past and future glacier recession in a changing climate. The main conclusions of this study are as follows:

- The MIE of the Late Holocene was reached when temperatures were 0.75°C to 1°C lower than the baseline climate period (1960-1990) under a calibrated melt rate factor.

- Currently, glaciated area ice thickness has retreated by 15% (low-end melt rate) to 20% (calibrated melt rate) compared to the MIE of the Late Holocene.

- Glacier mass loss is projected to occur at an unprecedented rate within the Late Holocene. Future simulations for 2070-2080 indicate a retreat more than double (-56±6%) compared to the ice loss from the MIE of the Late Holocene to the present.

- The glaciated area is expected to disappear towards 2090-2100.

Results confirm the ongoing imbalance of Eastern Nuussuaq, Central-Western Greenland GICs and highlight the unprecedented nature of current glacier shrinkage within the Late Holocene. Projections suggest that climate change will accelerate ice loss beyond historical trends, transforming Arctic landscapes, increasing deglaciated areas, and promoting the formation of new lakes. These findings enhance our understanding of Arctic peripheral glacier responses to anthropogenic climate change, with broad implications for hydrological and ecological systems."